# A meta-analysis of transcriptomic profiles of Huntington's disease patients

**Manuel Seefelder**[ID]\*, **Stefan Kochanek**

Department of Gene Therapy, Ulm University, Ulm, Germany

\* manuel.seefelder@uni-ulm.de

**Data Availability Statement:** All analyzed transcriptomic data are accessible from the Gene Ontology Omnibus (GEO) database of the National Center for Biotechnology Information (NCBI) using the accession numbers GSE33000, GSE129473, GSE64810, GSE1751, GSE24250, GSE8762,

## Abstract

Description of robust transcriptomic alterations in Huntington's disease is essential to identify targets for biochemical studies and drug development. We analysed publicly available transcriptome data from the brain and blood of 220 HD patients and 241 healthy controls and identified 737 and 661 genes with robustly altered mRNA levels in the brain and blood of HD patients, respectively. In the brain, a subnetwork of 320 genes strongly correlated with HD and was enriched in transport-related genes. Bioinformatical analysis of this subnetwork highlighted CDC42, PAK1, YWHAH, NFY, DLX1, HMGN3, and PRMT3. Moreover, we found that CREB1 can regulate 78.0% of genes whose mRNA levels correlated with HD in the blood of patients. Alterations in protein transport, metabolism, transcriptional regulation, and CDC42-mediated functions are likely central features of HD. Further our data substantiate the role of transcriptional regulators that have not been reported in the context of HD (e.g. DLX1, HMGN3 and PRMT3) and strongly suggest dysregulation of NFY and its target genes across tissues. A large proportion of the identified genes such as CDC42 were also altered in Parkinson's (PD) and Alzheimer's disease (AD). The observed dysregulation of CDC42 and YWHAH in samples from HD, AD and PD patients indicates that those genes and their upstream regulators may be interesting therapeutic targets.

## 1. Introduction

Huntingtin (HTT) functions in diverse cellular processes such as autophagy, endocytosis, vesicle transport, and transcriptional regulation [1]. A triplet repeat expansion in exon 1 of the HTT gene results in the expansion of an N-terminal polyglutamine tract and causes Huntington's disease (HD) [2]. Clinically, a progressive loss of motor functions, cognitive impairment, and psychiatric symptoms such as depression and anxiety [3] characterises HD. Besides neurological symptoms, HD patients suffer from a plethora of non-neuronal symptoms such as cardiac failure, muscle atrophy, impaired glucose tolerance, osteoporosis, weight loss, and testicular atrophy [4].

Expansion of the N-terminal polyQ tract impairs the multi-faceted function of HTT and its interaction with numerous other proteins [5,6]. Mutant huntingtin (mHTT), for instance, induces the activation of microglia, leading to increased secretion of interleukin-1β (IL-1β), tumour necrosis factor-alpha (TNF) and increased levels of reactive oxygen and nitrogen species [7]. Tabrizi et al. and Fan and Raymond showed that mHTT impairs the glutamate uptake

GSE113929, and GSE19677. R scripts used for the analysis can be retrieved from the GitHub repository under the following URL: https://github.com/ma-seefelder/HD_meta_analysis.

**Funding:** We acknowledge funding of this work by the Deutsche Forschungsgemeinschaft (DFG, German Research Foundation – project number 412854449).

**Competing interests:** The authors have declared that no competing interests exist.

in astrocytes leading to excitotoxicity [8,9]. Aberrant splicing of the mHTT mRNA results in the formation of a truncated HTT exon-1 protein forming nuclear and cytoplasmic inclusions [10]. R6/2 mice with a knock-in of exon-1 of human HTT show a more severe disease progression than mouse models with a knock-in of full-length mutant HTT [10]. The study of several huntingtin-interaction partners and their impaired function in HD further suggested impaired trafficking of clathrin-coated and non-coated vesicles in HD patients [11,12]. Transcriptomic studies of HD patients, cell lines, and mouse models expressing mHTT observed transcriptional dysregulation of a plethora of genes [13–19] such as differential regulation of genes involved in neuronal differentiation [14,16], heat shock response [13], mRNA processing [20], immune response and neuroinflammation [14]. Several mechanisms behind the broad transcriptional dysregulation such as altered expression of enhancer RNAs [21], sequestration of transcription factors (e.g. CREB1, TBP, or mSin3a) [22–25], or the sequestration of proteins such as the muscleblind-like splicing regulator 1 (MBNL1), nucleolin, and proteins of the small interfering RNA (siRNA) machinery [26,27] have been discussed.

Previously published analyses of transcriptomic profiles from HD patients [13–19] yielded varying results. Since a thorough knowledge of pathological mechanisms behind HD is essential for the design of further biochemical studies and development of therapies, we performed a meta-analysis of publicly available transcriptomic data from HD patients to identify genes altered in several studies. Within our meta-analysis, we found 661 and 737 genes with robustly altered mRNA levels in the blood and brain of HD patients, respectively. Strongly suggesting that dysfunction in protein transport and metabolism are central in HD, we identified by weighted gene co-expression network analysis a subnetwork of 320 genes, enriched in genes functioning in protein transport that strongly correlated with HD in the brain. Additionally, we identified the cell division cycle 42 (CDC42), p21 (CDC42 / RAC1) Activated Kinase 1 (PAK1), 14-3-3 protein eta (YWHAH), and protein phosphatase-2 catalytic subunit α (PP2CA) as hub genes of this subnetwork. Transcription factor enrichment analysis (TFEA) highlighted distal-less homeobox 1 (DLX1), high mobility group nucleosomal binding domain 3 (HMGN3), and protein arginine methyltransferase 3 (PRMT3) in this subnetwork. A signature of 74 and 41 genes, including CDC42 and YWHAH, were also altered in the brain of PD and in AD and PD patients, respectively. Similarly, a subnetwork of 118 genes, including genes coding for constituents of the Arp2/Arp3 complex, were significantly altered in the blood of HD patients. Strikingly, 78.0% of the genes in this blood subnetwork were direct or indirect targets of CREB1.

## 2. Results

### 2.1. Transcriptional changes in the brain of HD patients

#### 2.1.1. Identification of WGCNA modules correlating with HD in the human brain.
Since neurological and neuropsychiatric symptoms are the pathognomonic features of HD and possess a high disease burden for HD patients, several transcriptomic studies investigated transcriptional changes in the brain of HD patients. In this meta-analysis, we included three published transcriptomic studies using post-mortem brain tissue from the prefrontal cortex (NCBI accession number: GSE33000 and GSE64810) [14,16] and the caudate nucleus of prodromal HD patients (NCBI accession number GSE129473) [13] (Table 1). As described above, this meta-analysis aimed at identifying promising candidates for further functional studies and improving our understanding of transcription factors and mechanism, which are mainly affected in the brain of HD patients.

To identify genes with significantly altered mRNA levels in those three studies, we determined differentially expressed genes for each study separately, ranked them after their absolute

**Table 1. Information on patients from the analysed transcriptomic studies.**

| Accession number | No. HD patients | No. controls | Age (HD) | Age (Control) | Tissue | Reference |
|---|---|---|---|---|---|---|
| GSE33000 | 157 | 157 | 55.9 ± 14.32 | 63.5 ± 19.40 | DLPFC (BA9) | [16] |
| GSE129473 | 11 | 5 | 61.0 ± 21.59 | 62.8 ± 27.43 | Caudate nucleus / DLPFC (BA9) | [13] |
| GSE64810 | 20 | 49 | 58.25 ± 10.36 | 68.35 ± 15.83 | DLPFC (BA9) | [14,15] |
| GSE24250 | 8 | 6 | NA | NA | Venous cellular whole blood | [17] |
| GSE8762 | 12 | 10 | 48.4 ± 11.76 | 50.08 ± 8.63 | Lymphocytes | [18] |
| GSE1751 | 12 | 14 | NA | NA | Venous whole blood | [19] |

DLPC: Dorsolateral prefrontal cortex; BA: Broadmann area.

Z-ratio and performed a robust rank aggregation analysis (RRA). Thereby, we identified 737 differentially expressed genes (RRA score < 0.05) that were among the most altered genes in the analysed datasets (S1 File).

Based on all genes identified by RRA (S1 File), we performed a weighted gene co-expression analysis (WGCNA) to identify gene modules, i.e. clusters of highly correlated genes. Adjacency and the topological overlap matrix (TOM) for the gene network were calculated with a soft-thresholding power β of 14.5 for which the WGCNA network satisfies the criterion of scale-free topology ($R^2$ = 0.85) (S2 File). By module-based clustering with the diagonal, varying volume, and shape model (VVI), we identified nine modules of which the module eigengenes (first principal component) of seven modules (black, blue, red, brown, magenta, and turquoise) statistically significantly correlated with disease state (HD patients versus healthy individuals) as determined by correlation analysis (Fig 1A). Corroborating a potential link between genes in the black, brown and turquoise modules with HD, we found a positive and statistically significant correlation between module membership, defined as the probability that a gene belongs to this module, and gene significance, defined as the correlation between the expression values and the trait of interest, of 0.46 (p = 0.004), 0.47 (p = 1e-04), and 0.41 (p = 0.008), respectively (Fig 1A). Additionally, genes belonging to the black and brown module showed a high mean gene significance (Fig 1B). In contrast, indicating that transcriptomic alterations of genes of the turquoise modules were less pronounced than the genes of the black and brown module, we observed a low mean gene significance of genes of this module. Corroborating the results of our network analysis, we found that most of the identified genes or proteins are known to interact with several other proteins belonging to the same module. For instance, according to the network analysis in GeneMania, 94.05% and 94.11% of the genes are known to be co-expressed in humans. Likewise, network analysis using the STRING database (confidence cut-off = 0.4) [28] showed that 34 of 53 proteins (64.2%) of the black and 36 of 81 proteins (44.4%) of the brown module interact with at least one other protein.

Based on the clustering analysis of the modules eigengenes (Fig 1C) and the similarity of the eigengene adjacency (Fig 1D), we grouped the observed modules in three meta-modules: the first meta-module (M1) consisted of the black, blue, magenta, and red module, the second meta-module (M2) consisted of the brown, green and turquoise module, and the third meta-module (M3) consisted of the yellow and pink module. Combining the identified WGCNA modules to meta-modules and subsequent analysis of this meta-modules demonstrated a high correlation with HD (correlation r = 0.5, p-value = 1e-38), a positive correlation between gene significance and module membership of 0.73 (p-value = 2e-54), and the highest mean gene significance of all meta-modules for M1 (Fig 2A and 2B). Corroborating the importance of genes belonging to M1, the eigengene and adjacency of M1 clustered together with the disease state (Fig 2C and 2D). According to STRING, protein-protein interactions were strongly enriched

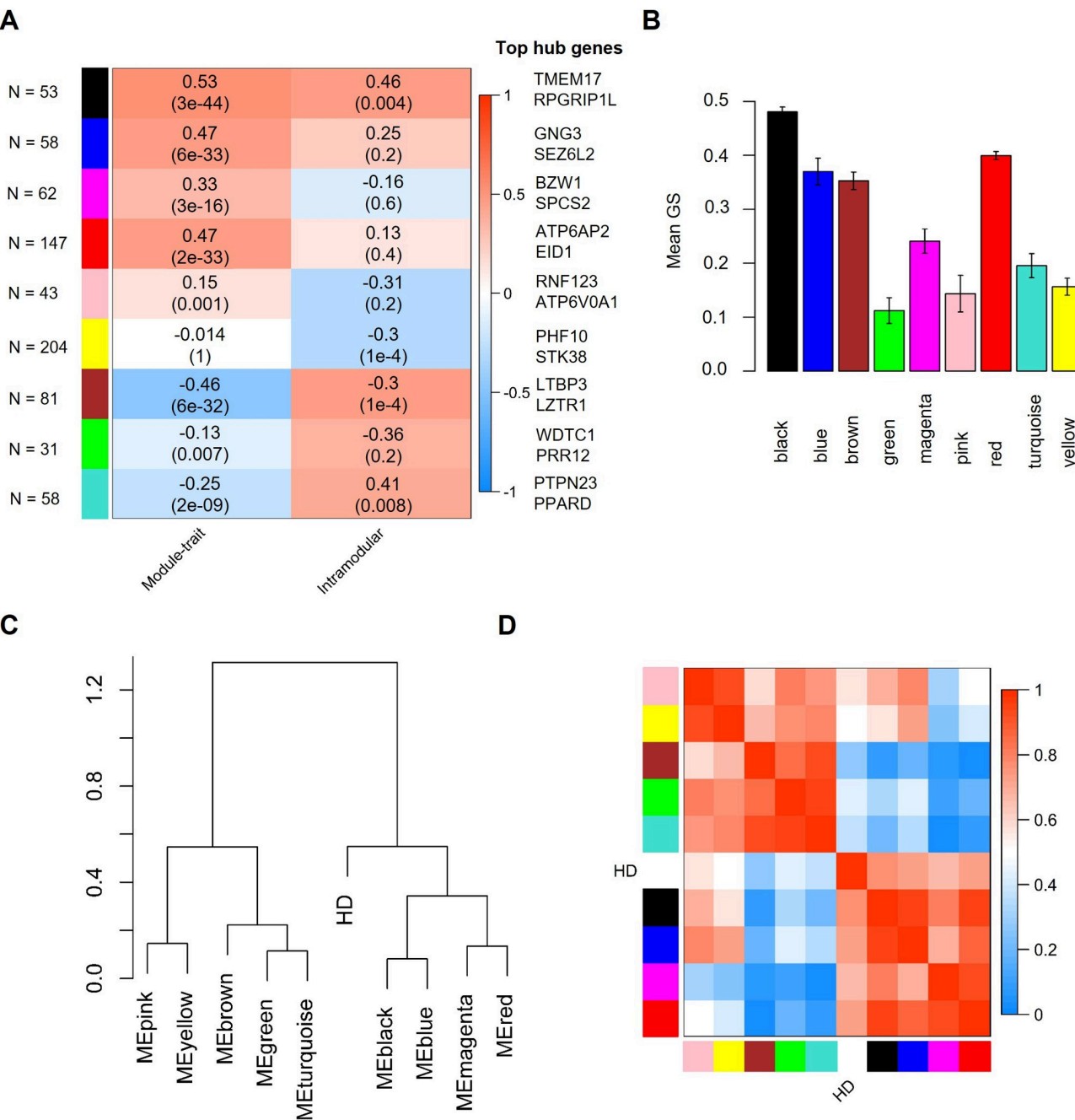

**Fig 1. Result of WGCNA analysis and the association of the disease state with module eigengenes.** A. Heatmap showing the correlation between module and disease state or between gene significance and module membership. A positive correlation between a module and disease state shows that the mRNA levels of genes belonging to this module were elevated in samples from HD patients and vice versa. P-values adjusted after Benjamini & Yekutieli (Yekutieli & Benjamini, 2001) are given in brackets. B: Mean gene significance of each module. Error bars depict the 95% confidence interval. C. Dendrogram showing hierarchical clustering of module eigengenes. D. Eigengene adjacency heatmap.

in M1 (826 observed edges, 602 expected edges, and p < 1.0e-16) at minimal interaction confidence of 0.4. While the eigengene of M2 correlated with HD, we could not observe a positive correlation between gene significance and module membership and, therefore, did not analyse this meta-module further.

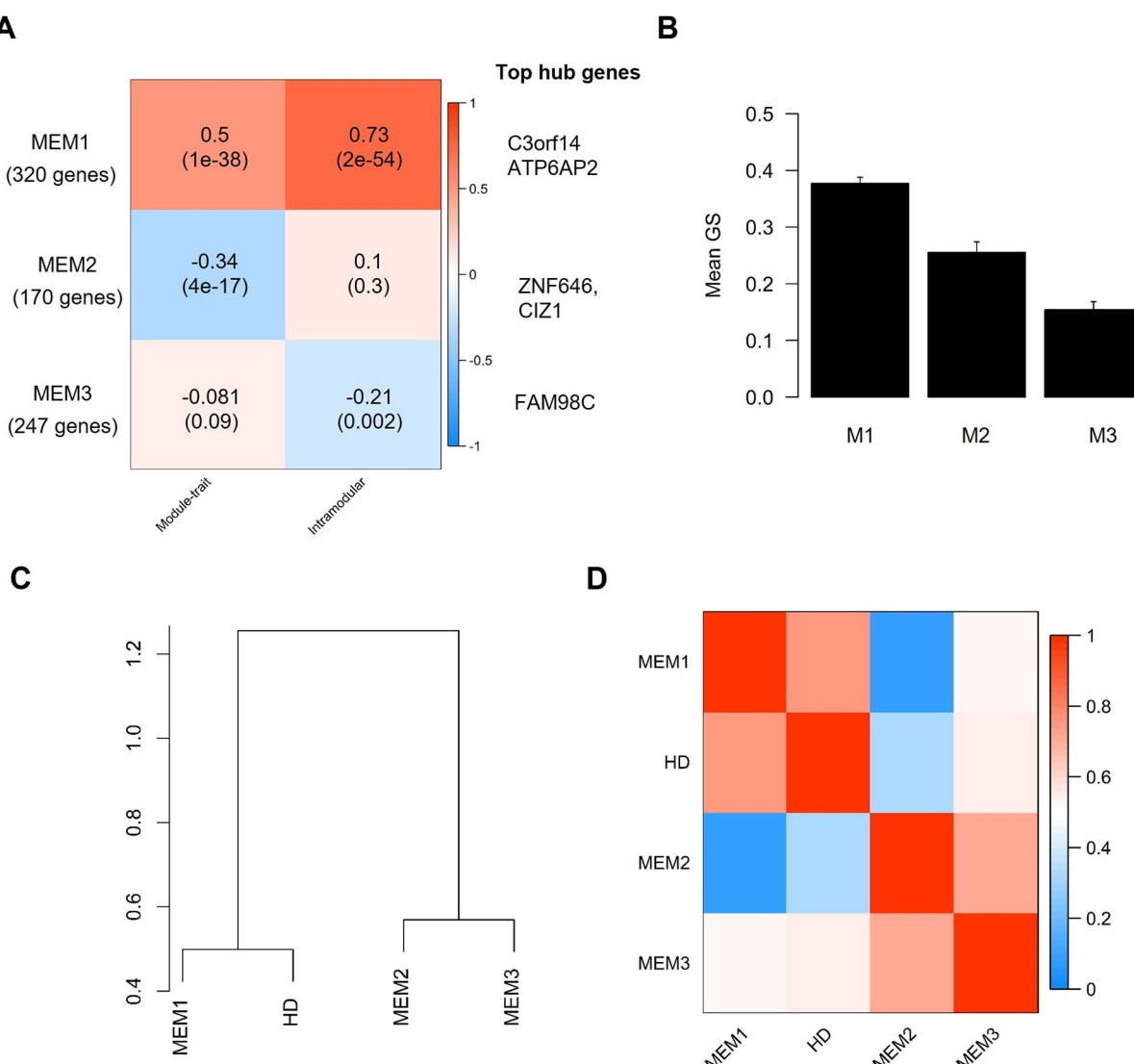

**Fig 2. Correlation of WGCNA meta-modules and the association of the disease state with module eigengenes.** A. Heatmap showing the correlation between the meta-module eigengenes (MEM1, MEM2, and MEM3), and disease state (HD) or between gene significance (GS) and meta-module membership. A positive correlation between a module and disease state shows that the mRNA levels of genes belonging to this module were elevated in samples from HD patients and vice versa. P-values adjusted after Benjamini & Yekutieli (Yekutieli & Benjamini, 2001) are given in brackets. B: Mean gene significance (GS) of each meta-module. Error bars depict the 95% confidence interval. C. Dendrogram showing hierarchical clustering of module eigengenes. D. Eigengene adjacency heatmap.

Because M1 showed the highest mean gene significance (Fig 3B), correlation with HD and between module-membership and gene significance (Fig 3A–3D) of all meta-modules and because the gene significance and module membership of genes belonging to M2 and M3 did not positively and significantly correlate, we focused all subsequent analyses on M1.

**2.1.2. Enrichment of genes involved in protein transport and cellular metabolism in the M1 meta-module.** Next, we intended at analysing the biological function of genes belonging to M1 which may enable to predict the pathophysiological consequences of altered mRNA levels of those genes in HD. To this end, we performed enrichment analyses against the gene ontology (GO) and Reactome database. Using the GO database, we found an enrichment of genes involved in protein transport (GO: 0015031, 12.8% of genes in M1 and FDR = 0.046)

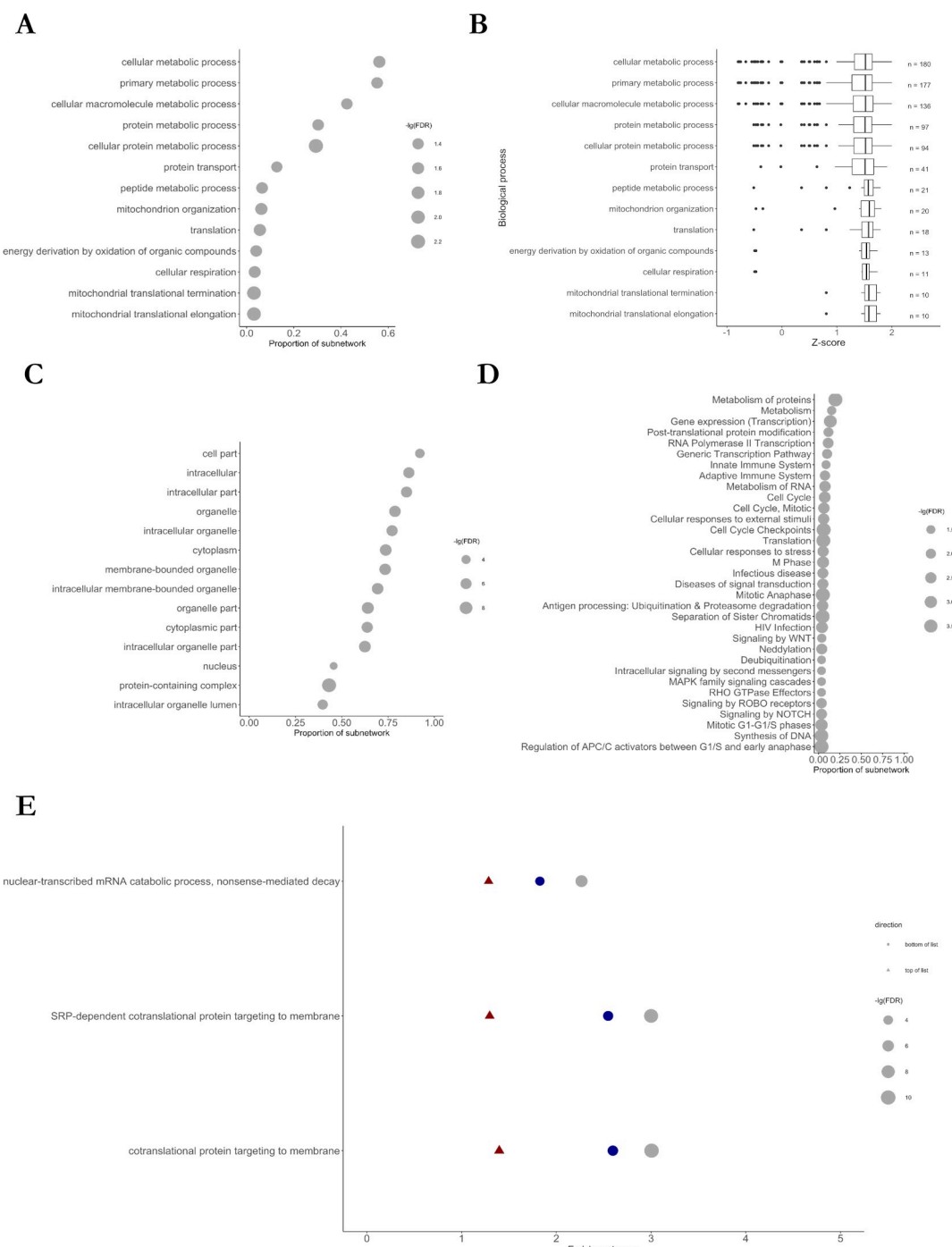

**Fig 3. Enrichment analysis of M1 subnetwork.** A: Gene ontology (GO) term enrichment analysis for biological processes. B: Mean Z-ratios of genes belonging to the enriched biological processes. C: Gene ontology (GO) term enrichment analysis for cellular compartments. D: Enrichment of Reactome pathways. The size of the circles depicts the negative decadic logarithm of the false-discovery rate (FDR). E: Result of gene set enrichment analysis against the gene ontology database. Only gene sets identified in each dataset (red: GSE33000; blue: GSE129473; grey: GSE64810) are shown and sets were ordered according to the score of the RRA analysis.

(Fig 3A) such as the Ras-related proteins 11A (RAB11A), 2A (RAB2A), 14 (RAB14), syntaxin-7 (STX7), syntaxin-12 (STX12), or the sorting nexin 3 (SNX3) in M1. Among the three datasets, the 41 proteins that belonged to M1 and function in protein transport processes showed a strong up-regulation in HD patient samples with a median Z-ratio of 1.51 (Fig 3B).

To also include genes with lower Z-ratios into the functional enrichment analyses, irrespective of their module membership, we additionally conducted gene set enrichment analysis (GSEA) of the three datasets independently. Strongly suggesting that the alteration of genes involved in protein transport may be relevant in HD, we found a strong enrichment of proteins involved in the co-translational protein targeting to the membrane in all three datasets by GSEA (Fig 3E). Additionally, genes functioning in cellular metabolic processes (GO:0044237, 56.3% of genes in M1 and FDR = 0.023), cellular respiration (GO:0045333, 3.4% of genes in M1 and FDR = 0.027) and translation (GO:0006412, 5.6% of genes in M1, and FDR = 0.023) were statistically significantly enriched in the M1 subnetwork. Similarly, when using the Reactome database, we observed a strong enrichment of proteins involved in protein metabolism, gene expression (transcription) and post-translational protein modification (Fig 3D).

Using the network enrichment analysis test (NEAT), we found a highly statistically significant over-enrichment of 38 KEGG pathways in M1 (S3 File). Among these enriched pathways were "SNARE interaction in vesicular transport" (adjusted p-value = 1.74e-33), "RNA transport" (adjusted p-value = 2.10E-217), and "mRNA surveillance pathway" (adjusted p-value = 9.63E-60) (S3 File).

**2.1.3. CDC42, PAK1, YWHAH and PPP2CA were identified as hubs in the M1 protein-protein interaction network.** Highly connected nodes (hubs) are defined as genes or proteins with significantly more connections with other nodes in the network. The functional relevance of hubs has previously been demonstrated. Increasing the confidence in identified hubs, we analysed whether hubs identified by WGCNA are also hubs in the protein-protein interaction network (constructed using STRING database) and co-expression networks.

As a hub gene of the WGCNA meta-module M1 with high intramodular connectivity, gene significance and module membership (Table 2 and S4 File), we identified CDC42 (Z-ratios: 1.72; 1.83; 1.45 in GSE33000, GSE129473 and GSE64810 respectively), a membrane-associated small GTPase that interacts with several effector proteins and thereby regulates cell migration [29], the bipolar attachment of spindle microtubules to kinetochores [30], the extension and

**Table 2. Top ten hub genes of meta-module M1.**

| Gene | IC | MM | GS | Z-ratio GSE33000 | Z ratio GSE129473 | Z ratio GSE64810 |
|---|---|---|---|---|---|---|
| C3orf14 | 60.22 | 0.96 | 0.48 | 2.01 | 0.46 | 1.77 |
| ATP6AP2 | 59.31 | 0.95 | 0.42 | 1.72 | 1.05 | 1.77 |
| ISCA1 | 57.93 | 0.95 | 0.47 | 1.96 | 0.75 | 1.77 |
| B3GALNT1 | 57.11 | 0.95 | 0.44 | 1.80 | 1.27 | 1.50 |
| POLR2K | 54.55 | 0.94 | 0.49 | 1.98 | 1.44 | 2.30 |
| PAK1 | 54.65 | 0.92 | 0.44 | 2.07 | - 0.42 | 0.30 |
| ACP1 | 53.75 | 0.94 | 0.45 | 1.82 | 1.22 | 1.76 |
| CDC42 | 53.60 | 0.94 | 0.42 | 1.72 | 1.83 | 1.45 |
| EID1 | 53.22 | 0.93 | 0.44 | 1.75 | 1.11 | 2.20 |
| RCN2 | 52.24 | 0.93 | 0.43 | 1.75 | 1.48 | 1.95 |

Hub genes were determined based on their intramodular connectivity, module membership and gene significance as described in the methods section. A positive Z-score shows upregulation in HD samples, while a negative Z ratio shows a downregulation. The complete list of identified hub genes can be retrieved from the supplement. IC: Intramodular connectivity; MM: Module membership; GS: Gene significance.

maintenance of the formation of filopodia, the dedicator of cytokinesis 10 (DOCK10) mediated spine formation [31], and the structural plasticity of dendritic spines [31]. Further corroborating the importance of CDC42 in the subnetwork correlating with HD, CDC42 was additionally central in the M1 protein-protein interaction network, constructed using the STRING database (S5 File). Together with CDC42, we identified 18 proteins with altered mRNA levels in all datasets that were directly connected with CDC42 according to the STRING database. Further, CDC42 interacts with other identified hub proteins such as the P21/Cdc42/Rac1-Activated Kinase 1 (PAK1) [32], 14-3-3 protein eta (YWHAH), or the protein phosphatase 2 catalytic subunit α (PPP2CA). mRNA levels of the CDC42 small effector 2 (CDC42SE2), that functions downstream of CDC42, was upregulated in the brain of HD patients (Z-ratios: 2.17; 0.97; 1.2). Additionally, CDC42 can interact with the CDC42-interacting protein 4 (CIP4), also known as thyroid hormone receptor interactor 10 (TRIP10) and HTT [33,34] that was not robustly dysregulated in our meta-analysis (Z-ratios -1.35, -0.03, 0.51). Besides the interaction of CIP4/TRIP10 with HTT and CDC42, CIP4/TRIP10 can interact with the vesicle-associated membrane protein 2 (VAMP2) and 7 (VAMP7) which are linked with other genes robustly altered in the brain of HD patients such as the vesicle-associated membrane protein 1 (VAMP1) and the Ras-related Protein Rab-14 (RAB14) [35].

In addition, we identified PAK1 (Z-ratios: 2.07; -0.42; 0.299), which can interact with CDC42, as a hub gene in the M1 subnetwork. Additionally, the PAK1 Interacting Protein 1 (PAK1IP1), which inhibits the activation of PAK1 by CDC42 through its interaction with the N-terminus of PAK1 (Xia et al, 2001), was upregulated in the brain of HD patients in all studies (Z-ratios: 1.88; 1.10; 1.98). Since PAK1, as well as PAK2 and PAK3, belong to the group A PAKs [36], we also analysed mRNA levels of other group A PAKs. While PAK2 was not robustly altered in the brain of HD patients (Z-ratios: -1.43; 1.31; 0.7), PAK3 mRNA levels (Z-ratios: 1.98; 0.46; 0.70) were slightly elevated in the brain of HD patients, although it was not identified by RRA (p = 0.10). Both CDC42 and PAK1 are interacting with another hub gene, the protein phosphatase 2 catalytic subunit alpha (PPP2CA) (Z-ratios: 1.98; 0.74; 1.91), which is an important phosphatase for microtubule-associated proteins. PPP2CA was additionally the most central protein in the network analysis of the M1 protein-protein interaction network (S5 File).

**2.1.4. DLX1, NFY, and PRMT3 target genes were enriched in the M1 subnetwork.** Broad transcriptional dysregulation in HD was often linked to direct interaction of mHTT with proteins of the small interfering RNA (siRNA) machinery [22–27] and different transcriptional regulators such as CREB1, TBP, mSin3a, MBNL1, nucleolin, histone deacetylases (HDACs), or the DNA methyltransferase 1 (DNMT1). Therefore, we performed a transcription factor enrichment analysis (TFEA) of the M1 subnetwork to define which transcription factors would best explain the observed alterations in the brain of HD patients. Analysis of the target genes of the top five TFEA hits, the mitochondrial transcription termination factor 3 (MTERF3), Myb/SANT DNA binding domain containing 4 with coiled-coils (MSANTD4), small nuclear RNA activating complex polypeptide 5 (SNPAC5), zinc finger protein 833 (ZNF833), and thymocyte nuclear protein 1 (THYN1), showed that most of their target genes were upregulated in the brain of HD patients (Fig 4 and S6 File). mRNA levels of MTERF3, MSANTD4, SNPAC5, ZNF833 and THYN1 were not consistently altered in the brain of HD patients. Alongside with the mRNA levels of their target genes, mRNA levels of the distal-less homeobox 1 (DLX1) (regulates 11.9% of M1, TFEA rank = 6, Z-ratios: 2.21; 0.16; 1.04), protein arginine methyltransferase 3 (PRMT3) (regulates 11.6% of M1, TFEA rank = 14, Z-ratios: 0.46; 2.65; 1.11), and nuclear transcription factor Y subunit β (NFYB) (regulates 29.7% of M1, TFEA rank = 24, Z-ratios: 1.05; 2.39; 2.02) were robustly upregulated in the brain of HD patients (Fig 4 and S6 File). Additionally, mRNA levels of the high mobility group nucleosomal binding

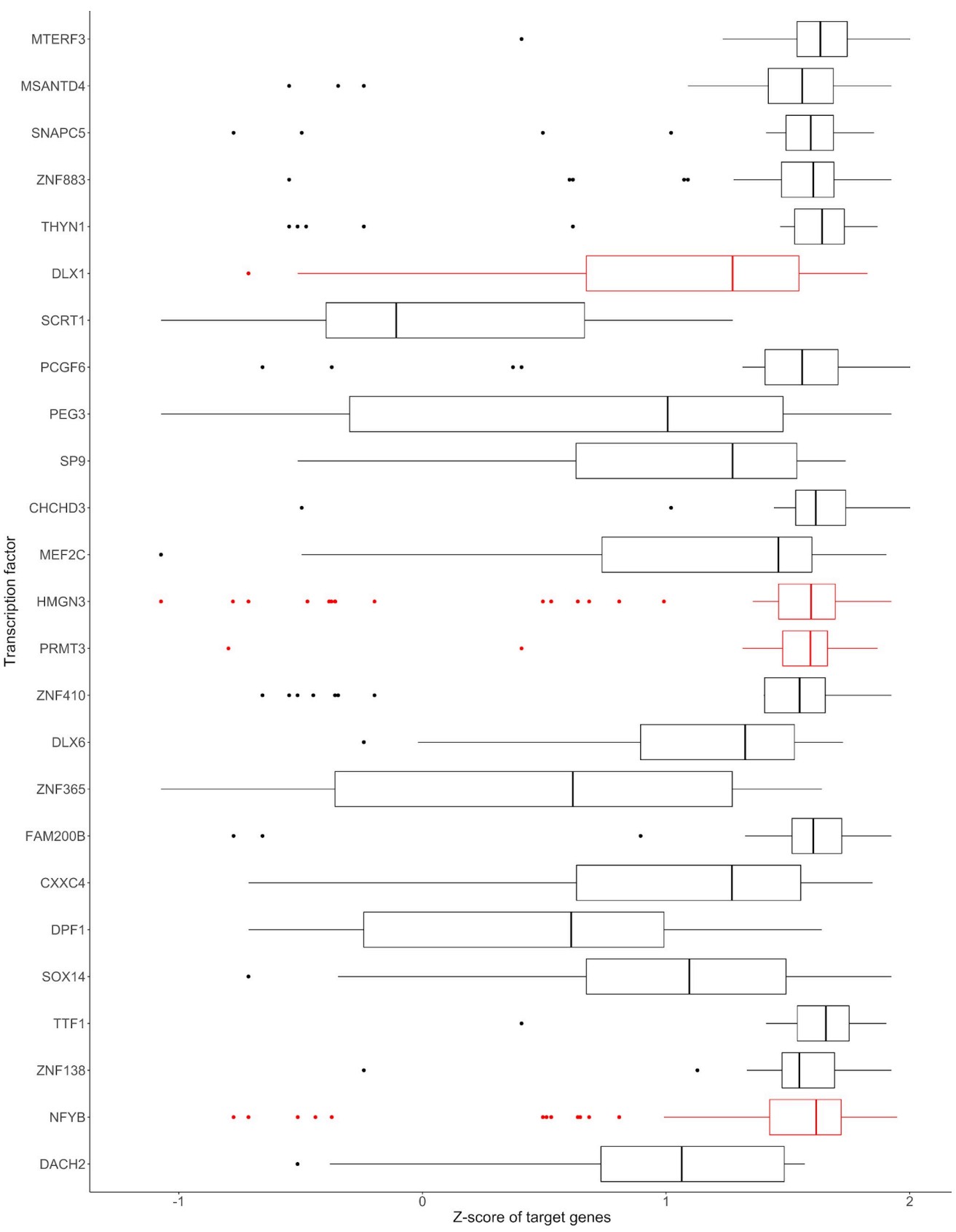

**Fig 4. Z-Scores of target genes from the top 25 enriched transcription factors.** Enrichment of transcription factors was computed with the Chea3 tool and the Z-ratios for each gene of M1 controlled by the respective transcription factor was averaged for the three different studies. Transcription factors were ordered based on the result of the transcription factor enrichment analysis with the best-ranked transcription factor at the top of the graph and transcription factors with robustly altered mRNA levels were depicted in red.

domain 3 (HMGN3) (regulates 24.1% of M1, TFEA rank = 13, Z-ratios: -0.91; 1.90; 1.77), that was additionally ranked high in the TFEA, appeared to be upregulated in only two of three studies and slightly downregulated in the other study (S1 and S6 Files). Strikingly, we noted that transcriptions factors that have previously been shown to be affected by mHTT such as CREB1 (rank 182) or TBP (rank 208) were ranked low or could not be detected at all (mSin3a) by TFEA.

Besides the effects of mHTT on transcription factors, previous publications indicated that the dysregulation of epigenetic modifiers such as DNMT1 or histone deacetylases (HDACs) (Federspiel et al, 2019; Siebzehnrübl et al, 2018; Moreno et al, 2016) might contribute to the broad transcriptional dysregulation in HD. In our meta-analysis, HDCA2 (Z-ratios: 0.69, 1.84, 1.86) and HDAC9 (Z-ratios: 2.26, 0.88, 1.42) were upregulated in the brain of HD patients, while the histone deacetylase 5 (HDAC5) mRNA levels were decreased (Z-ratios: -0.18; -2.17; -2.02). By RRA, we did not identify DNMT1, DNMT3A, or DNMT3B as robustly altered genes. Nonetheless, we noted a downregulation of DNMT1 (Z-ratio 0.48; - 1.35, -1.72) and DNMT3A (Z-ratio: 0.54, -1.38, -2.20) in datasets from Agus et al. 2019 and Labadorf et al., while both, DNMT1 and DNMT3A, were slightly upregulated in the larger dataset from Narayanan et al., 2014. DNMT3B (Z-ratio: - 0.76; -0.31; -0.79) was mostly unaltered in the brain of HD patients in all datasets.

**2.1.5. Summary of transcriptional changes in the brain of HD patients.** Taken together, mRNA levels of 320 genes of the M1 co-expression network strongly correlated with HD and genes involved in protein transport and metabolism was enriched in this co-expression sub-network (Fig 3A, 3B and 3D). CDC42, PAK1, YWHAH and PP2CA were identified as hub genes of this network. Especially substantiating on the relevance of CDC42, CDC42 has been indirectly linked with HD before [34] and can indirectly or directly interact with other identified hubs and 18 other proteins with robustly altered mRNA levels in the brain of HD patients. Further, the TFEA of the M1 subnetwork highlighted DLX1, NFY and HMGN3 as potential transcriptional regulators whose function might be affected in the brain of HD patients. mRNA levels of several epigenetic modifiers such as HDAC2, HDAC9, DNMT1, and DNMT3A were additionally altered in at least two of the three studies using HD brain samples.

## 2.2. A large proportion of differentially regulated genes in the brain of HD patients were also altered in Alzheimer's and Parkinson's disease

Previous transcriptomic studies have identified common transcriptional patterns between Alzheimer's disease (AD) and Parkinson's disease (PD) [37] and between AD and HD [16]. Hence, we compared the list of robustly altered genes in the brain of HD patients with the results of a previous meta-analyses comparing transcriptional alterations in PD and AD [37]. Of the 737 genes with robustly altered mRNA levels in the brain of HD patients, that were identified by RRA, 74 genes were also differentially expressed in PD and 41 genes were altered in all three neurodegenerative diseases (S7 File). Strikingly, alterations of mRNA levels of these genes were mostly reciprocal between HD and AD or PD, i.e. genes with an elevated mRNA level in the brain of HD patients showed decreased mRNA levels in the brain of AD or PD patients (S7 File).

Analysing of the co-expression networks demonstrated that 100% of these 74 genes were annotated as co-expressed in the GeneMania database (Fig 5A). 41 proteins whose mRNA levels were altered in HD and PD have at least one annotated interaction partner (Fig 5B). While PPP2CA, YWHAH, RAB11A and CDC42 which have been identified as hub genes/proteins in the M1 subnetwork network were also all central in the protein-protein interaction network (Fig 5B and S5 File) of genes differentially expressed upon HD and PD, only YWHAH was also central in the constructed co-expression network (Fig 5A and S5 File).

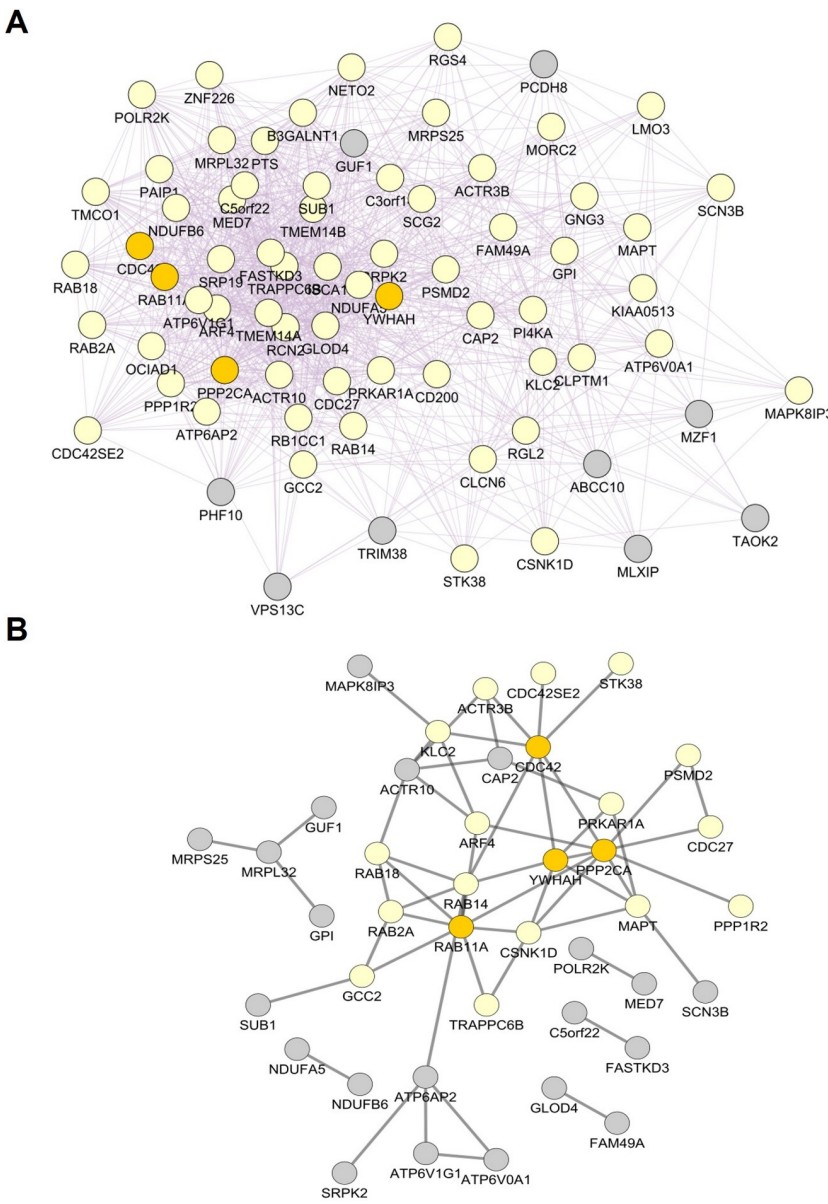

**Fig 5. Co-expression and protein-protein interaction network of genes altered in HD and PD. A.** Co-expression network. B. Protein-protein interaction network representing proteins that have at least one annotated interaction partner within the query. Networks were constructed with Cytoscape [38], its GeneMania [39] and STRING [28] plugins. A list of differentially regulated genes in the brain of PD patients was retrieved from [37]. Hub proteins of the identified M1 meta-module for HD patients and its direct connectors are depicted in dark orange and yellow, respectively.

## 2.3. Transcriptional changes in the blood of HD patients

As afore-described, HTT is ubiquitously expressed and HD symptoms are not confined to the central nervous system [1,4]. Hence, we additionally analysed transcriptomic studies of blood samples from HD patients and healthy controls (Table 1). Borovecki et al. 2005 (GSE1751) analysed the transcriptomic profile of twelve symptomatic and five presymptomatic HD patients in comparison to 14 healthy controls, whereas Hu et al. 2011 (GSE24520) included venous cellular whole blood samples from 6 healthy controls and 8 HD patients. Transcriptomic profiles of lymphocytes from 12 moderate stage HD patients and 10 age-matched healthy controls were analysed by Runne et al. 2007 (GSE8762).

**2.3.1. Identification of WGCNA modules correlating with HD in the human brain.** By robust rank aggregation (RRA), we identified 661 genes differentially expressed upon HD among the three datasets (p < 0.05) (S1 File). Based on those 661 genes with a soft-thresholding power β = 19.5 (scale-free topology R2 = 0.87) (S2 File) and subsequent module-based clustering with the diagonal, equal volume, varying shape (EVI) model, we identified nine WGCNA modules. Of these modules, the module eigengene of three modules (brown, pink, and yellow) statistically significantly correlated with the disease state (healthy individuals versus HD) (Fig 6A). While genes of the brown module showed a negative correlation between module membership and gene significance, genes of the pink and yellow modules showed a positive correlation (Fig 6A). Corroborating the importance of genes of the pink and yellow modules, genes of these modules showed the highest mean gene significance of all identified modules (Fig 6B). Further, 94.12% and 86.55% of the genes belonging to either the pink or yellow module were annotated by GeneMania to be co-expressed in humans. Owing to the low distance between the modules eigengenes of the pink and yellow modules (Fig 6C) and the highly similar adjacency of these modules (Fig 6D), we combined these modules for further downstream analysis and will further refer to this module as the blood meta-module (MB).

**2.3.2. Genes involved in transport and metabolic processes were enriched in the MB meta-module correlating with HD.** Like the enrichment of the meta-module M1 identified in the brain samples, we found a strong enrichment of proteins involved in transport (FDR = 0.03 and 35.6% of all genes in MB) and metabolic processes (FDR = 0.03 and 66.9% of all genes in MB) in the MB subnetwork (Fig 7A and 7B). Consistent with the enrichment of proteins involved in protein transport, we found a strong enrichment of proteins localised to endosome membranes (FDR = 0.002 and 9.3% of genes in MB) (Fig 7C). Among the 11 proteins localised to endosome membranes were the vesicle-associated membrane protein 7 (VAMP7) (Z-ratios -1.47; 0.99; 3.3), a paralog of VAMP2 that also displayed altered mRNA levels in the brain, and the sorting nexin 10 (SNX10) that is involved in membrane trafficking and protein sorting [40]. Additionally, we observed dysregulation of the actin-related proteins ACTR2/Arp2 (Z-ratio: -0.95; -1.75; 3.15), ACTR3/Arp3 (-1.92; -1.62; 2.78), and ARPC5 (-1; -1.87; 2.24) that together with ARBC1A, ARBC1B, ARPC3, and ARPC4 form a seven-subunit protein complex playing an essential role in the regulation of the actin cytoskeleton [41].

**2.3.3. The MB protein-protein interaction network and its network hubs.** By analysis of the MB protein-protein interaction network, ACTR2 was the most important hub protein, which strongly substantiates its pathophysiological relevance in HD. Further indicating that transcriptional dysregulation of regulators of the actin cytoskeleton may be important in HD, other actin-related proteins such as ACTR3B (Z-ratios: 1.3; 1.30; 1.62), ACTR6 (1.38; 1.54; 1.88) and ACTR10 (Z-ratios: 1.64; 1.23; 1.34) were robustly upregulated in the blood of HD patients according to this meta-analysis. The alteration of mRNA levels of constituents of the Arp2/3 complex and actin-related proteins further substantiates the identification of CDC42 as an important hub gene in the brain since CDC42 can activate the Arp2/3 complex through

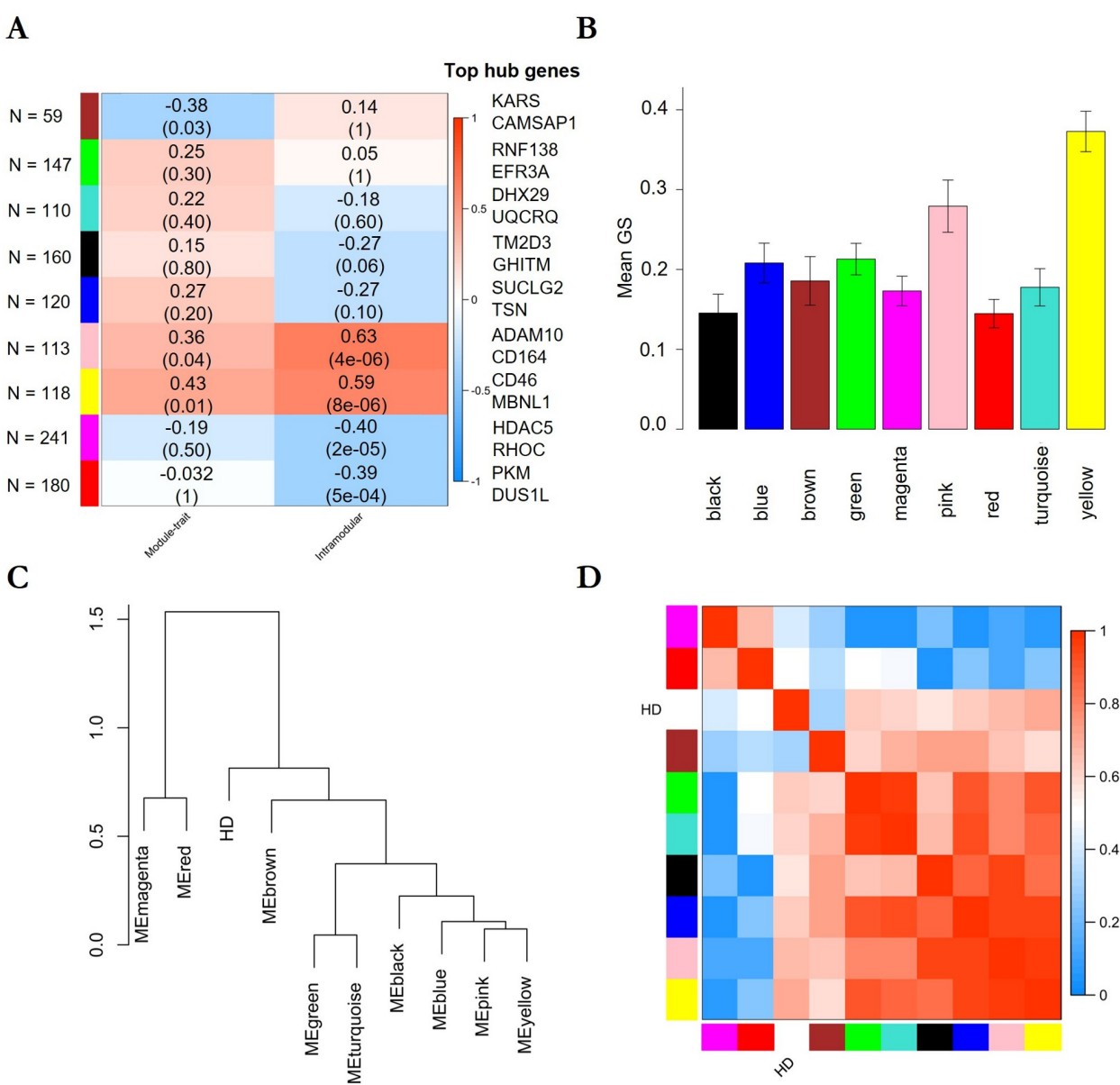

**Fig 6. Result of WGCNA analysis and the association of the disease state with module eigengenes.** A. Heatmap showing the correlation between module and disease state or between gene significance and module membership. A positive correlation between a module and disease state shows that the mRNA levels of genes belonging to this module were elevated in samples from HD patients and vice versa. P-values adjusted after Benjamini & Yekutieli (Yekutieli and Benjamini, 2001) are given in brackets. B: Mean gene significance of each module. Error bars depict the 95% confidence interval. C. Dendrogram showing hierarchical clustering of module eigengenes. D. Eigengene adjacency heatmap.

Wiskott-Aldrich syndrome proteins [42] such as WAS, WASF1, WASF2, WASF3, and WASL that appeared to be neither robustly altered in the brain nor the blood of HD patients.

**2.3.4. CREBP1 target genes were strongly enriched in MB.** To identify transcription factors that can regulate the transcription of genes belonging to MB, we performed a TFEA (S6 File). The CGG Triplet Repeat Binding Protein 1 (CGGBP1), zinc finger protein 654 (ZNF654), forkhead box N2 (FOXN2), and the specificity protein 3 transcription factor (SP3) were ranked at the top in the TFEA. Strikingly, transcription of these transcriptional regulators is regulated by the cAMP-responsive element-binding protein-1 (CREBP1). CREB1 was also

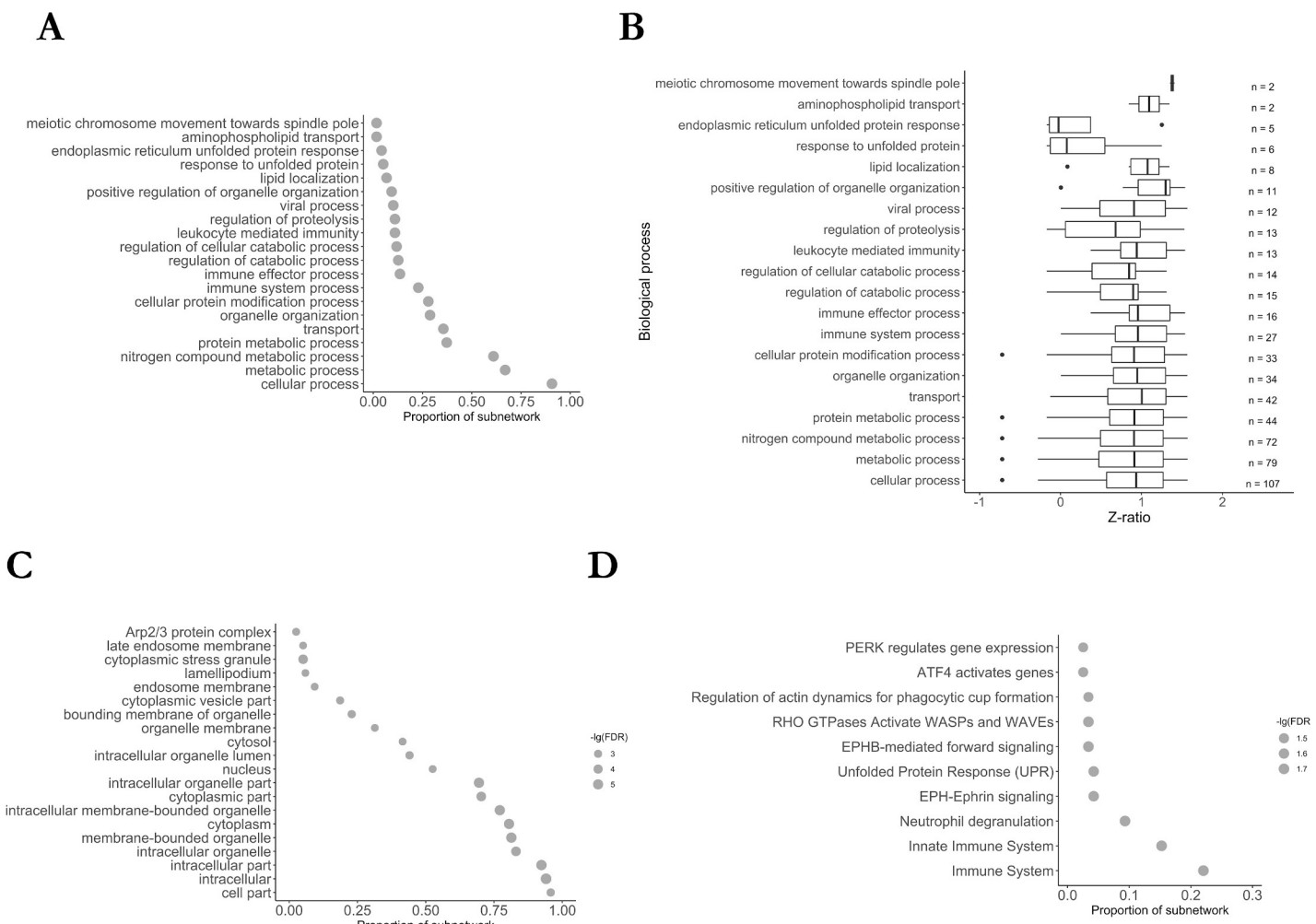

**Fig 7. Enrichment analysis of MB meta-module.** A: Gene ontology (GO) term enrichment analysis for biological processes. B: Mean z-ratios of genes belonging to the enriched biological processes. C: Gene ontology (GO) term enrichment analysis for cellular compartments. D: Enrichment of Reactome pathways.

ranked high (rank 15) in the TFEA and can regulate the transcription of genes coding for constituents of the Arp2/3 complex and hub genes of the MB subnetwork such as membrane-associated ring-CH-type finger 7 (MARCH 7), pumilio RNA binding family member 2 (PUM2), survival motor neuron domain containing 1 (SMNDC1), or zinc finger DHHC-type palmitoyltransferase 17 (ZDHHC17), also known as the huntingtin-interacting protein 14 (HIP14). CREB1 together with the other enriched transcription factors regulated by CREB1 (TET2, SP3, RLF, CGGBP1, ZNF148, FOXN2, ZNF654, ZBTB11, and ZNF770) regulates the transcription of 92 from 118 (78.0%) genes belonging to MB (S8 File).

Taken together, the enrichment of proteins localised to endosome membranes further corroborates the above-described alteration of protein-transport-related genes in the brain of HD patients. The dysregulation of several constituents of the Arp2/3 complex, which is activated by CDC42, substantiates the relevance of actin cytoskeleton dysregulation in HD.

## 2.4. 52 genes were differentially regulated in the blood and brain of HD patients

As noted before, the ubiquitous expression of HTT [1] and the clinical manifestation of HD outside the central nervous system [4] indicates that transcriptomic changes caused by polyQ

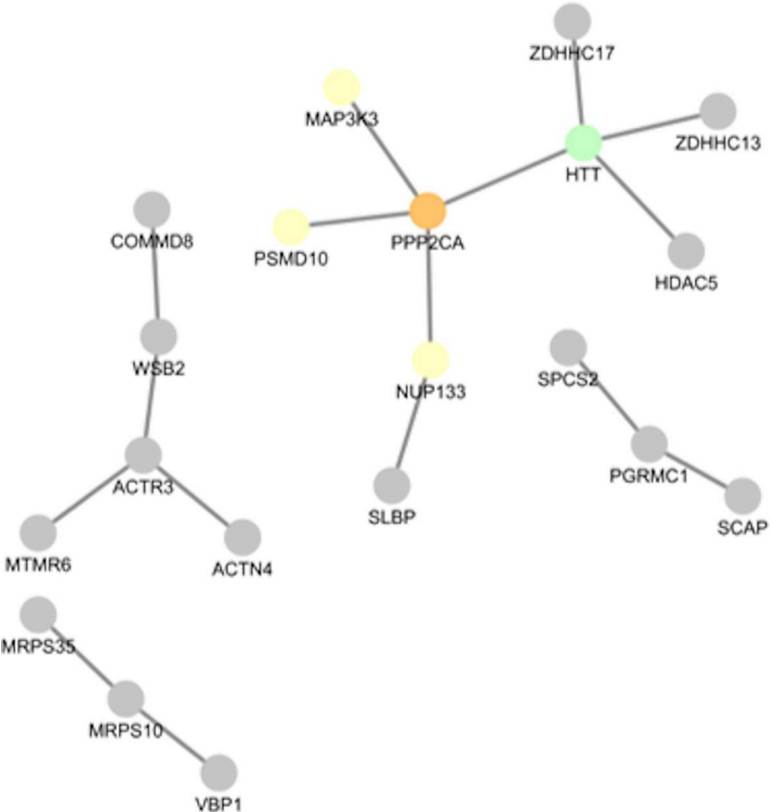

**Fig 8. Protein-protein interaction map of proteins with altered mRNA levels in the blood and brain of HD patients.** A protein-protein interaction map was constructed from the STRING database and plotted with Cytoscape. For illustration purposes, we added HTT (coloured in green) to the protein interaction map, although mRNA HTT were not robustly altered.

expansion of HTT may be not confined to nervous tissues. According to our meta-analysis, 52 genes were dysregulated in brain- and blood-derived samples from HD patients (S1 File).

Based on the genes/proteins altered in the blood and brain of HD patients, we constructed a protein-protein interaction network (Fig 8) to investigate the relationship between the identified genes. Furthermore, to identify transcription factors that might explain the observed transcriptomic alterations, we conducted a transcription factor enrichment analysis. While 82.4% of the proteins were annotated as co-expressed by GeneMania, 48.1% were interacting with a least one other protein according to the STRING databases (interaction cutoff: 0.4).

In our meta-analysis, we noted that mRNA levels of the zinc finger DHHC-Type palmitoyl-transferase 13 (ZDHHC13), i.e. huntingtin-interacting protein 14 (HIP14), and the zinc finger DHHC-Type palmitoyltransferase 17 (ZDHHC17), i.e. huntingtin-interacting protein 14-like protein (HIP14L) were mostly elevated in the blood and brain of HD patients. Previous studies linked the altered interaction between mHTT and ZDHHC17 and ZDHHC13 with altered regulation of the striatal N-Methyl-D-Aspartate Receptor (NMDA) trafficking [43].

TFEA analysis of the 52 genes altered in the blood and brain of HD samples highlighted the enrichment of NFY target genes (rank 3) (S6 File). NFY is a trimeric complex of proteins coded by the NFYA, NFYB, and NFYC genes. Besides the high ranking of NFYB in the combined dataset (rank 3), targets of NFYB were also enriched in the M1 (brain, rank 24) and the blood (rank 48) (S6 File). Moreover, NFYB (Z-ratios in the brain datasets: 1.05, 2.39, 2.02)

mRNA-levels and NFYB target genes (mean Z-ratio: 1.62) were also increased in the brain of HD patients (Fig 5 and S1 File).

## 2.5. HMGN3, NFY and CDC42 mRNA were additionally altered in the striatum of HD mice models and predictive for HD

To further substantiate the relevance of identified hub genes or transcriptional regulators such as CDC42, PAK1, YWHAH, DLX1, HMGN3, or NFY, we analysed transcriptomic alterations in the striatum of R6/2 [44] and YAC128 mice [45] (S9 File). Furthermore, we assessed how accurate control and HD mice can be discriminated based on mRNA levels of these genes (S10 and S11 Files). We also analysed transcriptomic data from the striatum of knock-in mice with different polyQ-lengths [46].

Indicating that NFY may play a role in HD across tissues, TFEA of the subnetworks M1 and MB and the combined dataset highlighted NFY. Like in human HD patients, NFYA (Z-ratios: 1.75, 0.77, 1.93 in R6/2, 12-month-old YAC128, and 24-month-old YAC128 mice respectively) and NFYB (Z-ratios: 1.31, 0.45, 1.38) mRNA levels were elevated in R6/2 and YAC128 mice. Moreover, NFYA mRNA levels were increased in the striatum of HD knock-in mice and correlated with polyQ-length and age (S9 File). Additionally, control and HD mice could be well discriminated based on NFYA (AUC = 0.86; 95% CI = [0.71, 1]), and NFYB (AUC = 0.79; 95% CI = [0.60, 0.98]) mRNA levels (S8–10).

In contrast, HMGN3 was only highlighted by TFEA of the M1 module, correlating with HD in the brain, and not in the blood datasets which may imply that dysregulation of HMGN3 may be confined to the brain In R6/2 and YAC128 mice, HMGN3 mRNA levels were elevated (Z-ratios: 1.89, 0.78, 2.06) in HD mice and control and HD mice could be well discriminated based on the HMGN3 mRNA levels. (AUC = 0.89; 95% CI = [0.76, 1]) (S 8–10). Furthermore, the increase of HMGN3 mRNA levels in the striatum of HD knock-in mice strongly correlated with polyQ length (S9 File). In concert with a more pronounced HD phenotype in 24-month YAC128 and R6/2 mice than in 12-month-old YAC128 mice, the increase of HMGN3, NFYA, and NFYB mRNA levels positively correlated with the age of YAC128. This raises the possibility that HMGN3, NFYA, and NFYB mRNA levels might be utilised as markers for disease progression and severity. However, further investigations on the usability of those genes as biomarkers in larger patient cohorts are required. Further, it should be clarified whether alteration of mRNA levels HMGN3, NFY, their target genes are specific for HD or whether they are present in other neurodegenerative diseases.

DLX1 and PRMT3 mRNA levels, which were both also highlighted by TFEA, were elevated in the striatum of R6/2 mice and HD knock-in mice but appeared to be unaffected in YAC128 mice. Like in the brain samples of HD patients, we observed a robust downregulation of DNMT3A (Z-ratios: - 1.24, -1.79, - 1.14) in the striatum of R6/2 and YAC128 mice and the mice could be discriminated based on DNMT3A levels (AUC = 0.83; 95% CI = [0.66, 0.99]). In disagreement with the analysis of brain samples from HD patients, DNMT1 and DNMT3B levels were not consistently altered in the striatum of R6/2 and YAC128 mice. Further corroborating the importance of CDC42 dysregulation in HD, CDC42 mRNA levels were elevated in the striatum of R6/2 and YAC128 (Z-ratios: 1.24, 1.18, 1.88). Additionally, control and HD mice could be discriminated based on CDC42 mRNA levels (AUC = 0.85; 95% CI = [0.68, 1]). mRNA level of PAK1, which was identified as a hub gene in the M1 subnetwork and correlated with HD, was merely elevated in YAC128 mice (Z-ratios: 1.47, 2.08), but was mostly unaffected in R6/2 mice (Z-ratio: - 0.45). The increase of CDC42 and PAK1 mRNA levels was also observed in HD knock-in mice expressing either Q140 or Q175 HTT (S9 File). While the increase in PAK1 mRNA levels correlated with the polyQ length in both 6- and 10-month-old

mice, the increase in CDC42 mRNA levels also slightly correlated with polyQ length in 10-month-old mice (S9 File).

## 3. Discussion

In our meta-analysis, we intended to identify by RRA, WGCNA and network analysis robust transcriptomic changes underlying HD. To this end, we included transcriptomic studies analysing different human brain regions and tissues from symptomatic and prodromal HD patients. Thereby, we identified subnetworks of 320 (M1) or 118 (MB) genes with robustly altered mRNA levels in the brain and blood of HD patients, respectively. Network analysis of differentially expressed genes in the brain highlighted CDC42, PAK1, YWHAH, and PP2CA as hub genes of the M1 subnetwork. Additionally, we identified a signature of 74 and 41 genes, including CDC42 and YWHAH, that were altered in the brain of PD and HD (Fig 5 and S7 File) and AD, PD and HD patients, respectively. In the blood, we identified a subnetwork of 118 genes, including genes coding for several constituents of the Arp2/3 complex that is activated by CDC42. TFEA highlighted the relevance of several already described (e.g. CREB1 and NFY) or novel (e.g. DLX1, PRMT3 and HMGN3) transcription factors that may play a role in HD. In conclusion, our analysis suggests that dysregulation of transcription factors and epigenetic modifiers, cellular metabolism, actin cytoskeleton and SNARE complex proteins play an important role in the pathology of HD (Fig 9).

As noted before, the pathology of HD is neither confined to certain brain regions nor the brain [4]. A successful HD therapy should, therefore, target a gene or protein that is not exclusively altered in a certain brain region or tissue. Hence, we analysed RNA data from different brain regions and blood samples in our meta-analysis, although an increased interstudy variability, reducing the sensitivity with which differentially expressed genes are identified, may argue against the combined analysis of different brain regions. Furthermore, certain limitations for the interpretation and combined analysis of transcriptomic data from different studies should be considered: although authors of the original publications strictly controlled RNA

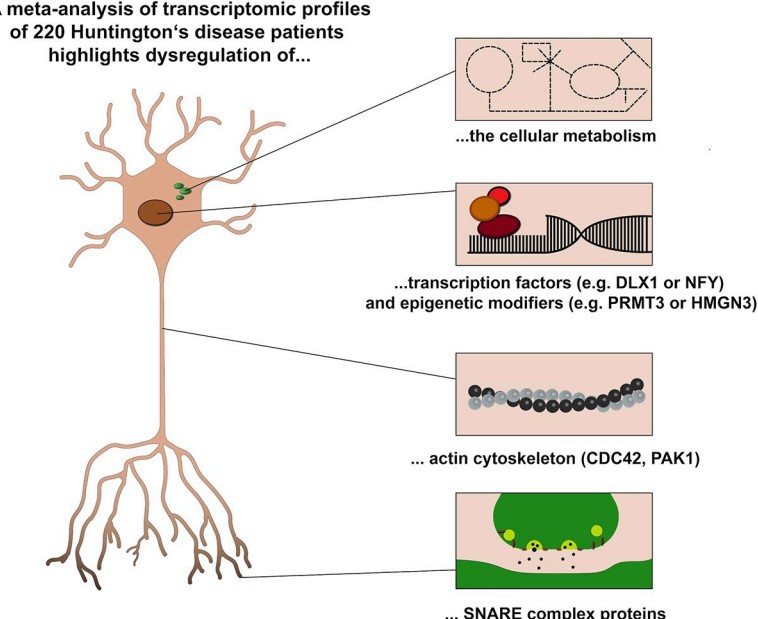

**Fig 9. Schematic illustration of transcriptomic alterations in the brain of HD patients.**

quality before RNA sequencing or microarray analysis, small changes in RNA quality might impair transcript quantification and subsequently also the results of this meta-analysis. Second, post-mortem samples from HD patients who died from HD can only provide insights into transcriptional changes at the end stage of HD that do not necessarily reflect changes at disease onset or during disease progression. Third, neurodegeneration in the brain of HD patients in late disease stages poses the risk that some of the observed alterations are caused by changes in tissue composition. Bearing the danger of altered tissue composition as a confounding factor in mind, we also included the dataset from Agus et al. 2019 who analysed early-stage, prodromal HD patients in which neuronal loss was less pronounced [13].

Several studies have shown that HTT and its interactors such as the huntingtin-associated protein 1 (HAP1) or the huntingtin-interacting protein 1 (HIP1) participate in protein transport and the organisation of the cytoskeleton [1]. RNAi-mediated silencing of the huntingtin-interacting protein 1 related (HIP1R), also known as huntingtin-interacting protein 12 (HIP12), for instance, led to the stable association of clathrin-coated structures and their endocytic cargo to dynamin, actin, the Arp2/3 complex, and cortactin [47,48]. Furthermore, HAP1 regulates synaptic vesicle exocytosis [49] and neuronal endocytosis through its interaction with the Sec23 homolog A, COPII coat complex component (SEC23A) and the clathrin light chain B [50]. In line with previous findings that HTT and its interactors regulate the cytoskeleton and transport processes and substantiating that impairments of these functions contribute to the pathophysiology of HD, we identified CDC42 as a hub gene in the M1 subnetwork that was highly correlated with HD in transcriptomic studies of post-mortem brain samples. CDC42 mRNA levels were also elevated in the brain of HD patients and the striatum of R6/2 and YAC128 mice. In contrast to HD, CDC42 mRNA levels were decreased in the brain of AD and PD patients [37]. Besides, mRNA levels of several constituents of the Arp2/Arp3 complex (ACTR2/Arp2: blood; ACTR3/Arp3: blood and brain; ACTR3B: brain; ARPC5: blood; ACTR6: brain), that interacts with the HIP1R-cortactin complex and is activated by CDC42, were altered in HD patients. mRNA levels of VAMP1, an indirect CDC42 interactor, were robustly upregulated in all studies using HD brain samples (S1 File). mRNA levels of VAMP2 and VAMP7 were additionally altered in the HD blood samples (S1 File), although these alterations were not consistent across the different studies. VAMPs are major constituents of protein complexes involved in the docking and fusion of vesicles [51]. These complexes are comprised of VAMPs, other syntaxins, the synaptosome associated protein 25 (SNAP-25), the N-ethylmaleimide-sensitive factor-like protein (NSF), the NSF-attachment proteins alpha (NAPA, SNAPA), beta (NAPB / SNAPB), gamma (NAPG / SNPAG), and SNAP receptors (SNARE). Corroborating our finding that the mRNA levels of VAMPs were altered in the brain and blood of HD patients and the striatum of R6/2 and YAC128 mice (this study), VAMP2 protein levels were also increased in striatal synaptosomes of Hdh140Q/140Q mice [52]. On the contrary, protein levels of other proteins involved in the docking and fusion of vesicles such as SNAP-25 or rabphilin 3a were reduced in the post-mortem cortex of HD patients [53]. Additionally, analysis of 175Q-HTT knock-in mice demonstrated altered levels of other proteins involved in synaptic function (SNAP-25, Rab3A, and PSD95), axonal transport, and microtubules (dynein, dynactin, and KIF3A) [54]. Additionally, HTT can interact with SNAP25, and the SNAP25-associated proteins syntaxin 1A (STX1A) and calcium voltage-gated channel auxiliary subunit alpha2delta 1 (CACNA2D1) [55]. Raising the possibility that HTT and its abundant interactor, the huntingtin-associated protein 40 (HAP40), plays a role in docking and fusion of synaptic vesicles, we previously found that constituents of this complex, the N-ethylmaleimide-sensitive factor attachment proteins alpha (NAPA, SNAPA), beta (NAPB / SNAPB), and gamma (NAPG / SNPAG) are the closest homologs of HAP40 [56].

Besides proteins of the SNARE complex, we identified other proteins linked with CDC42 in our network analysis of transcriptomic data from brain tissue of HD patients. For instance, PAK1, identified as a hub gene in the subnetwork M1, and its interactor PAK1IP1 showed robust upregulation in HD patients. The PAK proteins, PAK1 and PAK3, are central regulators of neuronal development and activating PAK1 mutations were aetiologic for secondary macrocephaly, developmental delay, ataxic gait and seizures in two unrelated patients [36]. Double knock-out of PAK1/PAK3 in mice affected brain size and structure [57]. Linking PAKs with HTT, Luo and Rubinszstein showed a physical interaction between HTT and PAK1 [58] and siRNA-mediated silencing of PAK1 and PAK2 reduced mutant HTT toxicity and aggregation [58,59]. PAK2 knock-down in the murine striatal cell line STHdh(Q111) also reduced mutant HTT toxicity [60].

As afore-mentioned, we found that 78% of the genes of the subnetwork MB, which strongly correlated with HD in the blood (Fig 6), were directly or indirectly regulated by CREB1 (S6 and S7 Files). CREB1, a leucine zipper transcription factor, activates the transcription of genes upon binding to the cAMP-response element (CRE). Steffan et al. previously showed that the CREB1-binding protein (CBP), a transcriptional coactivator of CREB1, can interact with HTT [22]. In a transcriptomic study of subcutaneous adipose tissue obtained from HD patients, CREB1 target genes were enriched, and the CREB1 mRNA levels were significantly increased (McCourt et al, 2015). Substantiating the physiological importance of CREB1, the double knock-out of CREB-1 and the cAMP-responsive element modulator (CREM) in mice (Creb1$^{Nescre}$Crem$^{-/-}$) led to severe neuronal loss during brain development and perinatal death [61]. A conditional, postnatal knock-out of Creb1 and Crem, showed considerable atrophy in the striatum and hippocampus and a dystonic phenotype [61]. Indicating that the loss-of-function of CREB1 in mice can partly be compensated by CREM, neither the loss of CREB1 nor CREM alone induced neurodegeneration in mice [61]. While we also found a strong enrichment of CREB1 target genes in our MB subnetwork by TFEA, CREB1 mRNA levels appeared to be unaltered in all analysed datasets. As afore-mentioned, TFEA of the M1 subnetwork consisting of genes with robustly altered mRNA levels in the brain of HD patients did not highlight CREB1. The finding that CREB1 targets were not enriched in the M1 subnetwork (S6) in combination with the data from Mantamadiotis et al., raises the possibility that CREB1 function may be affected in the brain of HD patients, but the CREB1 dysfunction is compensated by CREM or other transcription factors. In contrast, CREB1 dysfunction might not be compensated outside the brain due to the lack of detectable CREM protein expression in blood cells [62]. Another transcription factor whose transcription is controlled by CREB1 and whose target genes were enriched in the MB subnetwork is SP3. The dual treatment of R6/2 mice with mithramycin, inhibiting SP3, and cystamine reduced the hypertrimethylation of histone H3 and extended their overall survival over 40% [63]. The alteration of the activity or mRNA levels of CREB1 might also partly explain observed alterations in mRNA levels of CDC42 and constituents of the Arp2/3 complex (this meta-analysis) since CREB can regulate their transcription.

The TFEA performed in this study also highlighted the enrichment of NFY target genes in the M1 and MB subnetwork as well as in the dataset of genes affected in both tissues. In the brain of HD patients, mRNA levels of NFYB and its target genes were elevated (Fig 5 and S8 File). Further corroborating that the dysregulation of NFY and its target genes may be important in HD, van Hagen et al. found an enrichment of NFY target genes among a gene cluster that was differentially expressed in rat PC12 cells expressing the exon 1 of human 74Q-HTT [64]. Aggregates of mutant HTT can interact in vitro and in the mouse brain with NFY and thereby reduce transcription of the NFY target gene HSP70 [65]. On the other hand, our meta-analysis may indicate that the transcriptional activity and transcription of NFY might be

increased in the brain of HD patients, R6/2 and YAC128 mice. With regards to the currently available data on the role of NFY in HD, we cannot exclude the possibility that elevated mRNA levels of NFY and its target genes are caused by a compensatory mechanism to restore NFY function. This scenario may explain why transcription of NFY targets was reduced in one study [65], while we observed increased transcription of NFY and its targets.

Besides NFY and CREB-1, TFEA (S6) of the M1 subnetwork of genes altered in the brain of HD patients highlighted DLX1, PRMT3, and HMGN3 that may be involved in astrocyte maturation [66]. To our knowledge, this is the first study that indicates a potential role of DLX1, PRMT3, and HMGN3 dysfunction in HD. As noted above, DLX1 mRNA levels were only upregulated in the brain of HD patients in two studies, while it appeared to be unaffected in the third study analysing the caudate nucleus of prodromal HD patients. Additionally, DLX-1 mRNA levels were elevated in the striatum of R6/2 mice, while it was unaltered in the striatum of YAC128 mice. Besides the role of DLX1 in the adult brain, DLX1 plays also an important role in brain development; DLX1, together with NOLZ-1 and DLX2, regulates the migration of striatal neurones to the dorsal or ventral striatum and the identity of striatal projection neurones [67]. Chen et al. also demonstrated that the knock-out of NOLZ-1, also known as zinc finger protein 503 (ZNF503), in mice led to an upregulation of DLX1/2 and an aberrant neuronal migration from the dorsal to the ventral striatum [67]. Demonstrating that elevated DLX1/ DLX2 levels were causative for the aberrant neuronal migration, restoration of the altered DLX1/DLX2 levels in NOLZ-1 knock-out mice rescued the aberrant neuronal migration [67]. A conditional DLX1 knock-out in cortical interneurons in mice reduced the excitatory input, fewer excitatory synapses and hypoplastic dendrites [68] which substantiated the relevance of DLX1 beyond the striatum. Additionally, DLX1 knock-down in interneurons enhanced dendritic growth through neuropilin-2 and PAK3 [69], which was also slightly upregulated in the brain of HD patients (Z-ratios: 1.98; 0.46; 0.70) according to this meta-analysis. PRMT3, a protein methyltransferase whose mRNA levels were elevated in the brain of HD patients and the striatum of R6/2 mice (this study) but neither in the blood nor the striatum of YAC128 mice, is essential for dendritic spine maturation in the rat hippocampus [70] and neuronal development [71]. Due to a more rapid disease progression and disease onset in R6/2 than YAC128 mice [72,73], the elevation of DLX-1 and PRMT3 mRNA levels in the striatum of R6/2 but not in YAC128 mice raises the possibility that the dysregulation of DLX-1 and PRMT3 occurs in later disease stages and is more pronounced upon expression of the HTT-exon1 fragment.

Moreover, we identified a gene signature of 74 and 41 genes that were altered in the brain of PD and HD (Fig 5 and S7 File) and AD, PD and HD patients, respectively. As noted, before these genes were mostly reciprocally altered in HD and AD/PD, i.e. genes that were upregulated upon HD were downregulated in AD and PD. Currently, we do not have an explanation for the reciprocal alteration of these genes between HD and AD/PD. Further studies investigating the difference in the pathomechanisms of these neurodegenerative disorders may lead to further insights into this striking observation.

## 4. Conclusion

Here, we identified, by RRA and WGCNA, subnetworks of 320 (M1) and 118 (MB) genes with robustly altered mRNA levels in the brain and blood of HD patients, resp. In the brain, CDC42, PAK1, YWHAH, and PP2CA were highlighted as hub genes of the M1 subnetwork (S4 File), which appears to be enriched in genes functioning in protein transport (Fig 3). We also identified a signature of 74 and 41 genes, including CDC42 and YWHAH, that were altered in the brain of PD and HD (Fig 5 and S7 File) and AD, PD and HD patients, respectively. In blood, we identified a subnetwork of 118 genes, including genes coding for several

constituents of the Arp2/3 complex that is activated by CDC42. TFEA (Fig 5 and S6 File) highlighted the relevance of CREB1 in the pathology of HD since the transcription of 78.0% of genes altered in the blood of HD patients were directly or indirectly regulated by CREB1. Furthermore, DLX1, PRMT3, HMGN3 and NFY target genes were enriched in the identified modules. HMGN3, NFYA, NFYB, and CDC42 mRNA levels were additionally altered in R6/2 and YAC128 mice (S9 File) and could be used to discriminate between control and HD mice (S10 and S11 Files). Indicating that the upregulation of DLX1 and PRMT3 transcription may occur in later disease stages, DLX-1 and PRMT3 mRNA levels were merely elevated in R6/2 mice but not in YAC128 mice that show a less severe HD phenotype than R6/2 mice.

Our results strongly suggest that abnormal protein transport, cytoskeletal organization, and transcriptional regulation might be central features in the pathophysiology of HD (Fig 9). Furthermore, our study substantiates the role of CDC42, previously identified HTT interactors (e.g. PAK1, and PAK2) and transcriptional regulators (e.g. CREB1 and NFY) which have been reported to be sequestered to mutant HTT aggregates. Most interestingly, our data indicate a potential pathophysiological role of DLX-1, HMGN3 and PRMT3 in HD that have not been reported before.

## 5. Methods

### 5.1. Retrieval and tiding of datasets

In our meta-analysis, we analysed transcriptomic studies that were published in a peer-reviewed journal and whose raw data were publicly available. Furthermore, we excluded transcriptomic studies with less than eight samples from HD patients. To analyse post-mortem brain tissue, we retrieved data from the Gene Ontology Omnibus (GEO) database of the National Center for Biotechnology Information (NCBI) with the accession number GSE33000 [16], GSE129473 [13], and GSE64810 [14]. For the analysis of blood samples from HD patients, raw data were retrieved from the GEO database with the accession numbers GSE1751 [19], GSE24250 [17], and GSE8762 [18]. If genes were measured by several probes, the average of all probes of the respective genes was used. In our meta-analysis, we excluded samples from presymptomatic HD patients due to a low patient number.

Missing data in the dataset GSE33000 were imputed by sequential and random hot-deck imputation as implement in the R-package VIM [74] since we assumed missing at random after graphical analysis of missing values by the R-function matrixplot (VIM package) [75]. We normalised raw transcript-levels by quantile-normalisation using the R function normalize.quantiles.robust from the package preprocessCore (Bolstad, 2019) and, afterwards, converted them into Z-Scores.

### 5.2. Robust rank aggregation analysis (RRA)

To obtain a list of robustly altered genes, we computed Z-ratios according to the method proposed by Cheadle et al., 2003 [76] and ranked them after their absolute Z-ratio. The sorted transcript lists were analysed with RRA, as implemented in the R package RobustRankAggreg [77]. RRA is a distribution-based and parameter-free method that detects genes ranked consistently better than expected for uncorrelated genes (null hypothesis) and computes a significance score based on a probabilistic model [77]. The used RRA algorithm previously showed higher robustness to outliers, noise, and errors than other rank aggregation methods [77].

We included transcripts with an RRA score < 0.05 in the further downstream analysis and performed clustering analysis and plotting of the heatmaps with the function heatmap.2 implemented in the R-package gplots (version 3.0.3) [78].

## 5.3. Generation of weighted correlation networks

For the weighted correlation network analysis (WGCNA), the signed co-expression networks were build using the R-package WGCNA [79]. Correlation between genes was computed by biweight midcorrelation [79] to compute adjacency matrices. Based on the scale-free criterion [80], we set the power parameter β and computed the topological overlap measure (TOM) and the corresponding dissimilarity matrices (1 −TOM). Genes were clustered by model-based clustering of the dissimilarity matrix as implemented by Scrucca et al. 2016 [81]. Correlation of module eigengenes with disease state and between gene significance and module membership were calculated by Pearson's product-moment correlation as implemented in R (Langfelder et al, 2008; R Core Team, 2020). We adjusted p-values for multiple testing with the method described by Yekutieli & Benjamini, 2001 [82]. Genes with a gene significance score above 0.3, module membership above 0.7, and intramodular connectivity that is larger than the 8th percentile of all genes were identified as hubs.

## 5.4. Enrichment analysis

Gene set enrichment analysis (GSEA) was conducted with the algorithm implemented in the STRING database [28] for each dataset separately and results were combined by ranking enriched terms after their enrichment scores and aggregation by RRA [77]. Gene ontology (GO) and Reactome term enrichment of genes from different subnetworks were performed with the algorithms implemented by the STRING database [28] in Cytoscape (version 3.7.2) [38].

Based on the pathway annotations in the KEGG database [83] and the protein-protein interaction data from STRING (version 11.0) [28], we performed a network enrichment analysis test (NEAT) [84] as implemented in the R package ´neat´. Transcription factor enrichment analysis (TFEA) was performed with ChIP-X Enrichment Analysis 3 (CheA3) [85] using the mean rank as the metric. Furthermore, we calculated the mean Z-ratios of transcription factor targets with R-scripting language [86] and plotted the results using ggplot2 [87].

## 5.5. Network analysis

Protein-protein interaction networks (PPIN) were retrieved from the STRING database [28] using a confidence level cut-off of 0.4 and the Cytoscape software (version 3.7.2) [38]. The top 50 hubs of the PPIN were computed with the cytoHubba plug-in [88] using the betweenness, bottleneck, closeness, clustering coefficient, degree, DMNC, EcCentrity, EPC, MCC, radiality and stress scoring methods. Results of different scoring methods were aggregated by RRA [77] to increase the robustness of the prediction. Gene coexpression networks were constructed with Cytoscape software (version 3.7.2) [38] and the GeneMania plugin [39].

For the network of genes altered in HD, AD, and PD only the top 10 (protein-protein-interaction network) hub proteins or top 20 (coexpression network) were used. We included hub proteins with an RRA score below 0.05 in the further analysis and combined the lists of hubs in the co-expression network, identified by WGNA, and the protein-protein interaction network by RRA [77].

## 5.6. Comparison of differentially expressed genes in the brain of AD, PD, and HD patients

The list of differentially expressed genes in the brain of AD and PD patients was retrieved from Kelly et al 2019 [37] and for HD all genes with an RRA score in the brain below 0.05 were used. Networks and hub proteins/genes were computed as described above.

### 5.7. Confirmation of hub genes in HD mouse models

We analysed two datasets with transcriptomic data of HD mouse models to confirm if the identified hub genes and transcriptional regulators were additionally altered in independent datasets. In these studies, transcriptomic alterations in the striatum of R6/2 [44] (NCBI accession number: GSE113929) or YAC128 (NCBI accession number: GSE19677) [45] were analysed. Since Becanovic et al. 2010 used YAC128 mice at the age of 12 and 24 months, we separated the samples according to the age of mice and analysed them as independent datasets. Data tidying and computation of Z-ratios were performed as described in the sections "retrieval and tiding of datasets" and "Robust rank aggregation analysis (RRA)".

Classification analysis of selected genes (ACTR2, ACTR3, ARPC5, CDC42, CREB1, DLX1, DNMT1, DNMT3A, DNMT3B, HDAC2, HDAC5, HMGN3, NFYA, NFYB, NFYC, PAK1, PRMT3, VAMP2, VAMP7, YWHAH, ZDHHC13, ZDHHC17), identified in the human datasets, was performed, as implemented in the R-package pROC [89], to compute the area under the curve (AUC) of the respective receiver-operator characteristics (ROC). 95% confidence intervals of AUCs were calculated by bootstrapping with 10,000 replicates and genes with confidence intervals for the AUC above 0.5 were considered capable to discriminate between control and HD mice since the classification model is statistically significantly better than a random classification model.

## Supporting information

**S1 Fig.**
(TIF)

**S1 File. Genes with robustly altered mRNA levels in the brain and blood of HD patients.**
(XLSX)

**S2 File. Selection of the soft-thresholding power β.**
(XLSX)

**S3 File. Result of neat analysis.**
(XLSX)

**S4 File. Identified WGNA hub genes.**
(XLSX)

**S5 File. Hubs of protein-protein interaction networks.**
(XLSX)

**S6 File. Results of transcription factor enrichment analysis.**
(XLSX)

**S7 File. List of genes altered upon HD, PD and AD.**
(XLSX)

**S8 File. CREB-1 transcription factor subnetwork.**
(PDF)

**S9 File. Alteration of mRNA levels in the striatum of R6/2 and YAC128 mice.**
(XLSX)

**S10 File. Results of ROC analysis of selected genes and transcriptional regulators.**
(XLSX)

**S11 File. ROC curves of selected hub genes and transcriptional regulators.**
(PDF)

## Acknowledgments

We thank Andreas Neueder for discussions on the results of this meta-analysis. Furthermore, we thank Robin Nilson for improving the design of Fig 9.

## Author Contributions

**Conceptualization:** Manuel Seefelder.

**Funding acquisition:** Stefan Kochanek.

**Investigation:** Manuel Seefelder.

**Methodology:** Manuel Seefelder.

**Validation:** Manuel Seefelder.

**Visualization:** Manuel Seefelder.

**Writing – original draft:** Manuel Seefelder, Stefan Kochanek.

**Writing – review & editing:** Manuel Seefelder, Stefan Kochanek.

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
