## [Decision Letter · Decision Letter 0]

18 Mar 2021

PONE-D-21-03462

A meta-analysis of transcriptomic profiles from Huntington’s disease patients points to a pathophysiological role of CDC42, NFY, DLX1 and PRMT3

PLOS ONE

Dear Dr. Seefelder,

Thank you for submitting your manuscript to PLOS ONE. After careful consideration, we feel that it has merit but does not fully meet PLOS ONE’s publication criteria as it currently stands. Therefore, we invite you to submit a revised version of the manuscript that addresses the points raised during the review process.

We look forward to receiving your revised manuscript.

Kind regards,

Xiao-Hong Lu, M.D., Ph.D.

Academic Editor

PLOS ONE

Journal Requirements:

Reviewers' comments:

Reviewer's Responses to Questions

**Comments to the Author**

1. Is the manuscript technically sound, and do the data support the conclusions?

Reviewer #1: Yes

Reviewer #2: Partly

2. Has the statistical analysis been performed appropriately and rigorously? 

Reviewer #1: Yes

Reviewer #2: I Don't Know

3. Have the authors made all data underlying the findings in their manuscript fully available?

Reviewer #1: Yes

Reviewer #2: Yes

4. Is the manuscript presented in an intelligible fashion and written in standard English?

Reviewer #1: Yes

Reviewer #2: Yes

5. Review Comments to the Author

Reviewer #1: Transcriptional deficit is a key mechanism in Huntington’s disease (HD) pathogenesis. There have been a fair number of RNA-seq studies in HD patients (brain tissues and blood), HD animal models and cell cultures. In this study, Seefelder and Kochanek performed meta-analysis on some of the public available RNA-seq datasets from HD patients, including applying robust rank aggregation (RRA) analysis to integrate and identify the top ranked dysregulated genes among several datasets, using WGCNA to identify disease-related module in these genes, and then annotate the modules with enrichment analysis. The authors identified new genes and transcriptional regulators that are dysregulated in HD and may also play a role in other neurodegenerative diseases.

This study appeared to present novel biological insights on HD transcriptopathy, and the authors put great effort to evaluate and review individual genes and the related pathways, for example, they have a thorough discussion to explain the discrepancy between HD brain and blood difference in CREB1 regulated pathways.

However, the quality of the data can be further improved and the authors need to address the following issues to substantiate their claims.

1. RRA analysis is the basis of such bioinformatic study on existing RNA-seq datasets, the authors need to show the RRA score < 0.05 is stringent enough for this study. For example, one of the top hits identified in this paper, PAK1, its expression varied a lot in 3 datasets used here (Z=2.07, -0.42, 0.30), while PAK3 (Z=1.98, 0.46, 0.70) was not identified by RRA, such results are puzzling. The authors need to explain whether they added any weights to certain dataset(s). Other similar researches often used RRA score <0.01 as the threshold, some even applied the leave one out cross-validation on the RRA algorithm to get more stable and trustworthy p-values.

2. The authors compared their finding in HD patients to two HD mouse models: R6/2 and YAC128. The latter is known for having very few transcriptional deficits, even at 2-yr old. People has performed a more sophisticated transcriptional study in HD knock-in mice series, and identified CAG-length and age-dependent gene expression deficits in multiple tissues (Langfelder et al., Nat Neurosci. 2016). This dataset should be used to confirm the key hub gene expressions (DLX-1, PRMT3 etc. ).

3. The quality of the figures need to be improved: Figure 1, 2 and 6 has minimal information, as the authors didn’t show how many genes in each module or meta-module, how they distribute among HD and control samples (using module eigengenes), which are the top genes. All these can and should be added to the figures. Also, in Figure 5 and 8 (String data), hub gene and its direct connected nodes (e.g. CDC42 and its connectors) can be coloured differently in STRING database, to make them more noticeable.

4. The authors mentioned “Strikingly, alterations of mRNA levels of these genes were mostly reciprocal between HD and AD or PD, i.e. genes with an elevated mRNA level in the brain of HD patients showed decreased mRNA levels in the brain of AD or PD patients”. Please add data and discussion about this point. Another paper by Moss et al. (Scientific Reports, 2017) performed RNA-seq in HD patients’ brains and blood, and showed HD shared some common signatures with AD. This dataset should be added to this study if possible, or at least be compared to the current data and discussed the similarities and differences.

Reviewer #2: Seefelder and Kochanek 2021

This is an interesting study analyzing the many publicly available gene expression datasets comparing Huntington’s disease patient post-mortem brain tissue and blood to healthy controls. The meta-analysis includes 220 HD patients and 241 healthy controls. The inclusion of all of these datasets allows a good assessment of gene expression changes across various studies. There should be more analyses of this type performed in order to provide a unifying picture to the HD field around gene expression alteration that does not require the generation of altogether new datasets.

Amongst other genes, they ultimately identified perturbations in a large number of genes involved in protein transport and highlighted some of the “hubs” of this subnetwork in the work described in the manuscript. They also highlight some of the genes identified as also altered in Parkinson’s and Alzheimer’s disease.

This work overall is sound and uses both blood and tissue. The inclusion of blood offers the potential to identify changes that may be tractable with disease progression. They use multiple platforms and analyses including, robust rank aggregation analysis to determine the differences in gene expression for each of the studies that was included in the meta-analysis and then performed WGCNA to identify the gene modules. They further analyzed these modules with additional models.

In general, there is likely considerable knowledge to be gained from this work. However, there is so many analyses performed here that the manuscript as written is hard to follow, thus considerable revision is needed. This work should be revised in terms of how the data is presented as far as 1) additional subheadings, 2) grouping of the analyses in a way that allows the reader to follow more clearly the approaches being taken and why they are taken (TFEA analyses and GO analyses, STRING databases, GeneMania, etc.). It should be clear why the M1 network is focused on in the work.

6. PLOS authors have the option to publish the peer review history of their article (what does this mean?). If published, this will include your full peer review and any attached files.

Reviewer #1: No

Reviewer #2: No

---

## [Author Response · Author response to Decision Letter 0]

7 Apr 2021

Dear Professor Lu, 

we thank you for considering our manuscript for publication in PLOS ONE and thank the two reviewers for their careful evaluation of the manuscript, their constructive critic, and their thoughtful suggestions.

In the following, we address the comments raised by you and the two reviewers in detail:

Editor / Editorial Office:

“Please ensure that your manuscript meets PLOS ONE's style requirements, including those for file naming.”

We checked again whether the manuscript meets all style requirements and made some changes in referencing the supporting information and file naming. 

"We note that you have included the phrase “data not shown” in your manuscript. Unfortunately, this does not meet our data sharing requirements. PLOS does not permit references to inaccessible data. 

We require that authors provide all relevant data within the paper, Supporting Information files, or in an acceptable, public repository. Please add a citation to support this phrase or upload the data that corresponds with these findings to a stable repository (such as Figshare or Dryad) and provide and URLs, DOIs, or accession numbers that may be used to access these data. Or, if the data are not a core part of the research being presented in your study, we ask that you remove the phrase that refers to these data.”

We have removed the phrase “data not shown” in lines 114 and 369 since the database containing the mentioned information has been already cited in the same sentence.

“ We note that you have indicated that data from this study are available upon request. PLOS only allows data to be available upon request if there are legal or ethical restrictions on sharing data publicly. For more information on unacceptable data access restrictions, please see http://journals.plos.org/plosone/s/data-availability#loc-unacceptable-data-access-restrictions.”

We have now uploaded the R code to a GitHub repository that is publicly accessible under the URL https://github.com/ma-seefelder/HD_meta_analysis. Henceforth, we would like to request a change in the statement on data availability from “R scripts used for the analysis are made available upon reasonable request to the corresponding author” to “ R scripts used for the analysis can be retrieved from the GitHub repository under the following URL https://github.com/ma-seefelder/HD_meta_analysis”. 

Reviewer 1

“RRA analysis is the basis of such bioinformatic study on existing RNA-seq datasets, the authors need to show the RRA score < 0.05 is stringent enough for this study. For example, one of the top hits identified in this paper, PAK1, its expression varied a lot in 3 datasets used here (Z=2.07, -0.42, 0.30), while PAK3 (Z=1.98, 0.46, 0.70) was not identified by RRA, such results are puzzling. The authors need to explain whether they added any weights to certain dataset(s). Other similar research often used RRA score <0.01 as the threshold, some even applied the leave one out cross-validation on the RRA algorithm to get more stable and trustworthy p-values.”

We agree with the reviewer’s comment that the selection of the RRA score is crucial for the results of any study using robust rank aggregation analysis. Concerning the reviewer’s recommendation of using leave-one-out cross-validation on the RRA algorithm, we believe that this cannot be applied here because only three different studies were compared. In our opinion, an RRA score threshold of 0.05 is the optimal compromise between stringency and sensitivity. An RRA score threshold of 0.05 has been also previously frequently used by others (Song et al. Aging, 2019; Gholaminejad et al., Heart Failure Reviews, 2021; Zare et al., Int J Mol Cell Med., 2019; Jun A et al., Cancers, 2021; Zhang et al., J Cell Mol Med, 2021). Moreover and importantly, we applied RRA only for the initial screening of genes with altered mRNA levels and combined it with WGCNA analyses. The subsequent WGCNA analysis would further exclude genes that did not correlate with HD. 

Concerning the “puzzling result” of PAK1 and PAK3, it should be noted that the RRA score of PAK1 is at the edge of significance with a p-value of 0.049, whereas the PAK3 only has a slightly higher RRA score of 0.1. Further, the presented results of altered PAK1 mRNA levels in YAC128 (Z-ratios: 1.47, 2.08) further substantiates the likely relevance of alterations in PAK1 mRNA levels in PAK1. The extensive discussion of PAK1 and other identified hits in the manuscript was not only based on the robust rank aggregation analysis but also the other bioinformatical analyses (WGCNA, protein interaction networks, murine data). 

“The authors compared their finding in HD patients to two HD mouse models: R6/2 and YAC128. The latter is known for having very few transcriptional deficits, even at 2-yr old. People have performed a more sophisticated transcriptional study in HD knock-in mice series and identified CAG-length and age-dependent gene expression deficits in multiple tissues (Langfelder et al., Nat Neurosci. 2016). This dataset should be used to confirm the key hub gene expressions (DLX-1, PRMT3 etc.).”

Following the reviewer’s suggestion, we now have added this dataset to our meta-analysis and added the corresponding data to supplementary file 9. In these datasets, the change in mRNA levels of NFYA, HMGN3, and PAK1 strongly correlate with the polyQ length of the knocked-in HTT gene. CDC42 mRNA levels were also elevated in these mice but the correlation with the polyQ length was less and only visible in 10-month old mice (line 399)

“The quality of the figures need to be improved: Figure 1, 2 and 6 has minimal information, as the authors didn’t show how many genes in each module or meta-module, how they distribute among HD and control samples (using module eigengenes), which are the top genes. All these can and should be added to the figures. Also, in Figure 5 and 8 (String data), hub gene and its directly connected nodes (e.g. CDC42 and its connectors) can be coloured differently in STRING database, to make them more noticeable.”

Based on the reviewer’s comments we have modified figures 1, 2, and 6 and have added the number of genes and the two hub genes with the highest intramodular connectivity for each module. Concerning the reviewer’s comment on the distribution among HD and control samples, we have added a further explanatory sentence to the caption describing that a positive correlation score in the heatmap shows that mRNA levels of genes from the respective module are elevated in samples from HD patients. For figure 5 and 8, we coloured the hub genes and their direct connectors differently. 

“The authors mentioned “Strikingly, alterations of mRNA levels of these genes were mostly reciprocal between HD and AD or PD, i.e. genes with an elevated mRNA level in the brain of HD patients showed decreased mRNA levels in the brain of AD or PD patients”. Please add data and discussion about this point. 

Data to this comment have been added to the supplementary files (S7). Based on the reviewer’s r comment, an additional reference to the supplementary table containing the data was added. Moreover, we added a paragraph discussing the observation (lines 588).

Another paper by Moss et al. (Scientific Reports, 2017) performed RNA-seq in HD patients’ brains and blood and showed HD shared some common signatures with AD. This dataset should be added to this study if possible, or at least be compared to the current data and discussed the similarities and differences.”

Moss et al. 2017 (Scientific Reports, doi: 10.1038/srep44849) described that in an Alzheimer’s disease brain the transcriptional signature overlapped with a transcriptional signature that they identified in the blood of HD patients. In contrast to the present meta-analysis, Moss et al. did not compare transcriptional signatures in the brain of HD patients with changes in the brain of AD patients. As described in the manuscript, we did not observe a large overlap of transcriptional changes between blood and brain of HD patients. 

We believe, that the comparison of transcriptional signatures between AD and HD in the study of Moss et al. and this meta-analysis would not lead to significant insights into the transcriptopathy of HD or other neurodegenerative diseases in the brain. 

Reviewer 2

“However, there are so many analyses performed here that the manuscript as written is hard to follow, thus considerable revision is needed. This work should be revised in terms of how the data is presented as far as 1) additional subheadings, 2) grouping of the analyses in a way that allows the reader to follow more clearly the approaches being taken and why they are taken (TFEA analyses and GO analyses, STRING databases, GeneMania, etc.).”

Based on the reviewer’s comment, we have added l subheadings in the manuscript to improve its clarity and structure (e.g. lines 71, 140f., 172f., 215, 255, 266, 291f., 299, 321). Further, we now explain the reasons behind each bioinformatical approach in more detail (e.g. lines 142ff. and 174ff.) 

We believe that in this revision the more detailed structure of the manuscript and additional explanations for the reasons behind different approaches strongly improved the comprehensibility of the manuscript. 

"It should be clear why the M1 network is focused on in the work"

We added a more concise explanation of why the M1 network is focused on the work (lines 136ff.). The main criterium for our decision was the absence of a positive and significant correlation between gene significance (GS) and module membership (MM) of genes belonging to M2 and M3 as described in lines 115ff and figure 2A. A negative correlation between GS and MM, as for M3 indicates that genes that are more central in the respective WGCNA module are less correlated with, in this case, HD. 

We believe that we have addressed all comments of the Reviewers. We would like to resubmit this revised version of the manuscript for further evaluation.

Sincerely,

Manuel Seefelder and Stefan Kochanek

---

## [Decision Letter · Decision Letter 1]

4 May 2021

PONE-D-21-03462R1

A meta-analysis of transcriptomic profiles from Huntington’s disease patients points to a pathophysiological role of CDC42, NFY, DLX1 and PRMT3

PLOS ONE

Dear Dr. Seefelder,

Thank you for submitting your manuscript to PLOS ONE. After careful consideration, we feel that it has merit but does not fully meet PLOS ONE’s publication criteria as it currently stands. Therefore, we invite you to submit a revised version of the manuscript that addresses the points raised during the review process.

We look forward to receiving your revised manuscript.

Kind regards,

Xiao-Hong Lu, M.D., Ph.D.

Academic Editor

PLOS ONE

Journal Requirements:

Reviewers' comments:

Reviewer's Responses to Questions

**Comments to the Author**

The authors have addressed most of the concerns, but there are two remaining minor issues: 1. Please include gene numbers and top hub genes for each modules in Fig 1a, just like in Fig 2 and 6. 2. The authors identified and discussed quite a few genes and related pathways in the manuscript, but selected 4 genes to put in the title. A general title without gene names or just emphasizing CDC42 is more suitable.

---

## [Author Response · Author response to Decision Letter 1]

25 May 2021

Dear Professor Lu, 

we thank you for considering our manuscript for publication in PLOS ONE and thank the two reviewers for their careful evaluation of the manuscript, their constructive critic, and their thoughtful suggestions.

In the following, we address the comments raised by you and the two reviewers in detail:

Please include gene numbers and top hub genes for each module in Fig 1a, just like in Fig 2 and 6. 2. 

 Figure 1 has been also updated with the information on gene numbers and hub genes in each module. 

The authors identified and discussed quite a few genes and related pathways in the manuscript, but selected 4 genes to put in the title. A general title without gene names or just emphasizing CDC42 is more suitable.

 The title of the manuscript was changed in response to your comment. We decided on the more general title “A meta-analysis of transcriptomic profiles of Huntington’s disease patients”.

We believe that we have addressed all comments of the Reviewers. We would like to resubmit this revised version of the manuscript for further evaluation.

Sincerely,

Manuel Seefelder and Stefan Kochanek

---

## [Editor Report · Decision Letter 2]

28 May 2021

A meta-analysis of transcriptomic profiles of Huntington’s disease patients

PONE-D-21-03462R2

Dear Dr. Seefelder,

We’re pleased to inform you that your manuscript has been judged scientifically suitable for publication and will be formally accepted for publication once it meets all outstanding technical requirements.

Kind regards,

Xiao-Hong Lu, M.D., Ph.D.

Academic Editor

PLOS ONE
---

## [Editor Report · Acceptance letter]

2 Jun 2021

PONE-D-21-03462R2 

A meta-analysis of transcriptomic profiles of Huntington’s disease patients 

Dear Dr. Seefelder:

I'm pleased to inform you that your manuscript has been deemed suitable for publication in PLOS ONE. Congratulations! Your manuscript is now with our production department. 

Kind regards, 

on behalf of

Dr. Xiao-Hong Lu 

Academic Editor

PLOS ONE